# IMEX-Reg: Implicit-Explicit Regularization in the Function Space for Continual Learning

**Prashant Bhat[1],*, Bharath Renjith[1],*, Elahe Arani[1,2],†, Bahram Zonooz[1],†**
**[1]Eindhoven University of Technology (TU/e)    [2]Wayve**
`{p.s.bhat, e.arani, b.zonooz}@tue.nl, bharathcrenjith@gmail.com`

Reviewed on OpenReview: `https://openreview.net/forum?id=p1a6ruIZCT`

## Abstract

Continual learning (CL) remains one of the long-standing challenges for deep neural networks due to catastrophic forgetting of previously acquired knowledge. Although rehearsal-based approaches have been fairly successful in mitigating catastrophic forgetting, they suffer from overfitting on buffered samples and prior information loss, hindering generalization under low-buffer regimes. Inspired by how humans learn using strong inductive biases, we propose **IMEX-Reg** to improve the generalization performance of experience rehearsal in CL under low buffer regimes. Specifically, we employ a two-pronged implicit-explicit regularization approach using contrastive representation learning (CRL) and consistency regularization. To further leverage the global relationship between representations learned using CRL, we propose a regularization strategy to guide the classifier toward the activation correlations in the unit hypersphere of the CRL. Our results show that IMEX-Reg significantly improves generalization performance and outperforms rehearsal-based approaches in several CL scenarios. It is also robust to natural and adversarial corruptions with less task-recency bias. Additionally, we provide theoretical insights to support our design decisions further. [1]

## 1 Introduction

Deep neural networks (DNNs) deployed in the real world frequently encounter dynamic data streams and must learn sequentially as the data becomes increasingly accessible over time (Parisi et al., 2019). However, continual learning (CL) over a sequence of tasks causes catastrophic forgetting (McCloskey & Cohen, 1989), a phenomenon in which acquiring new information disrupts consolidated knowledge, and in the worst case, the previously acquired information is completely forgotten. Rehearsal-based approaches (Aljundi et al., 2019; Buzzega et al., 2020; Arani et al., 2022) that maintain a bounded memory buffer to store and replay samples from previous tasks have been fairly effective in mitigating catastrophic forgetting. In practice, however, the buffer size is often limited due to memory constraints (such as on edge devices) and in longer task sequences due to restricted sample-to-task ratios. In such low buffer regimes, repeated learning on bounded memory drastically reduces the ability of CL models to approximate past behavior, resulting in overfitting on buffered samples (Bhat et al., 2022a), exacerbated representation drift at the task boundary (Jeeveswaran et al., 2023) and prior information loss (Zhang et al., 2020), impeding generalization across tasks.

Humans, on the other hand, exhibit a remarkable ability to consolidate and transfer knowledge between distinct contexts in ever-changing environments (Barnett & Ceci, 2002), rarely interfering with consolidated knowledge (French, 1999). In the brain, CL is mediated by a plethora of neurophysiological processes that harbor strong inductive biases to encourage learning generalizable features, which require minimal adaptation when encountered with novel tasks. On the contrary, due to the lack of good inductive biases, DNNs often latch onto patterns that are only representative of the statistics of the training data (Sinz et al., 2019).

---

*Equal contribution.    †Equal advisory role.
[1]Code is available at: `https://github.com/NeurAI-Lab/IMEX-Reg`.

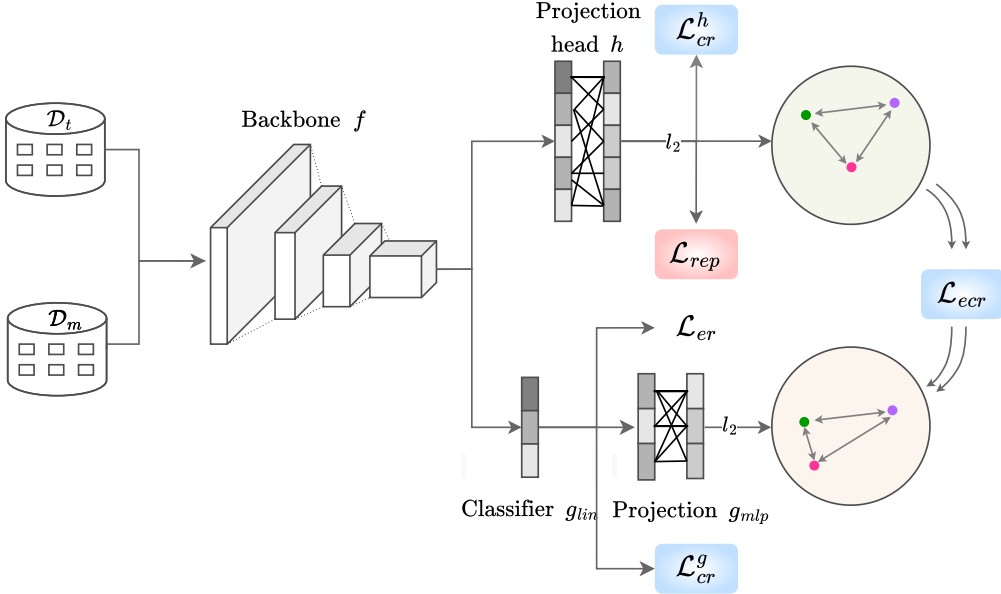

Figure 1: Implicit - Explicit Regularization in CL: IMEX-Reg employs CRL ($\mathcal{L}_{rep}$) and consistency regularization ($\mathcal{L}_{cr}^{g}$ and $\mathcal{L}_{cr}^{h}$) to bias the learning towards generalization. To further leverage desirable traits of learning on unit-hypersphere using CRL, IMEX-Reg aligns the geometric structures within the classifier projection's hypersphere with that of the projection head's hypersphere ($\mathcal{L}_{ecr}$) thereby compensating for the weak supervision under low-buffer regimes.

Consequently, DNNs are typically susceptible to changes in the input distribution (Koh et al., 2021; Hendrycks et al., 2021). Therefore, leveraging inductive biases to incorporate prior knowledge in DNN can bias the learning process toward generalization.

Regularization has traditionally been used to introduce inductive bias in DNNs to prefer some hypotheses over others and promote generalization (Ruder, 2017). Multitask learning (MTL), a form of inductive transfer that involves learning auxiliary tasks, acts as an implicit regularizer by introducing an inductive bias without imposing explicit constraints on the learning objective (Ruder, 2017). Sharing representations between related tasks in MTL helps DNNs to generalize better on the original task (Caruana, 1997). Moreover, assuming that the tasks in MTL share a common hypothesis class, sharing representations across tasks primarily benefits tasks with limited training samples (Liu et al., 2019). Contrastive representation learning (CRL) (Chen et al., 2020b; Khosla et al., 2020) as an auxiliary task in MTL promotes generalization in the shared parameters by maximizing the similarity between positive pairs and minimizing the similarity between negative pairs. A vast number of unsupervised CRL methods learn representations with a unit-norm constraint, effectively restricting the output space to the unit hypersphere (e.g. He et al. (2020)). Intuitively, having the features live on the unit hypersphere leads to several desirable traits: Fixed-norm vectors are known to improve training stability in modern machine learning where dot products are ubiquitous (Xu & Durrett, 2018). Moreover, if features of a class are sufficiently well clustered, they are linearly separable with the rest of the feature space (Wang & Isola, 2020). In addition to implicit regularization using CRL as an auxiliary task, the aforementioned desirable characteristics can be further leveraged for explicit classifier regularization, thereby compensating for weak supervision under low buffer regimes.

We propose IMEX-Reg, a two-pronged CL approach aimed at implicit regularization using hard parameter sharing and multi-task learning, and an explicit regularization in the function space to guide the optimization of the CL model towards generalization. As CRL captures the global relationship between samples using instance discrimination tasks, we seek to align the geometric structures within the classifier hypersphere with those of the projection head hypersphere to compensate for the weak supervision under low buffer regimes. Our contributions are as follows:

- Inspired by how humans leverage inductive biases, we propose IMEX-Reg, a two-pronged implicit (Khosla et al., 2020) - explicit (Arani et al., 2022) regularization approach to mitigate catastrophic forgetting in image classification in CL.
- As having the features lie on the unit-hypersphere leads to several desirable traits, we propose a regularization strategy to guide the classifier toward the activation correlations in the unit-hypersphere of the CRL.
- We show that IMEX-Reg significantly improves generalization performance and outperforms rehearsal-based approaches in mitigating catastrophic forgetting in CL. IMEX-Reg is robust to natural and adversarial corruptions and well-calibrated with less task-recency bias.
- We also provide theoretical insights on feature similarity between CRL and cross-entropy loss, and the Johnson-Lindenstrauss (JL) lemma (Dasgupta & Gupta, 2003) connecting our explicit regularization loss to support our design decisions better.

## 2 Related Works

**Rehearsal-based approaches:** Continual learning on a sequence of tasks with non-stationary data distributions results in catastrophic forgetting of older tasks, as training the CL model with new information interferes with previously consolidated knowledge (McClelland et al., 1995; Parisi et al., 2019). Experience-Rehearsal (ER) (Ratcliff, 1990; Robins, 1995) is one of the first works to address catastrophic forgetting by explicitly maintaining a memory buffer and interleaving previous task samples from the memory with the current task samples. Several works are built on top of ER to reduce catastrophic forgetting further in CL under low buffer regimes. The low-buffer regime is characterized by a small sample-to-task ratio, signifying limited data availability per task, while the high-buffer regime entails a large sample-to-task ratio, indicating ample data per task, influencing the model's ability to handle CL challenges. The low-buffer regime poses significant challenges for CL systems, such as higher susceptibility to catastrophic forgetting and the need for effective memory consolidation and utilization. Deep Retrieval and Imagination (DRI) (Wang et al., 2022a) uses a generative model to produce additional (imaginary) data based on limited memory. ER-ACE (Caccia et al., 2022) focuses on preserving learned representations from drastic adaptations by combating representation drift under low buffer regimes. Gradient Coreset Replay (GCR) (Tiwari et al., 2022) proposes maintaining a coreset to select and update the memory buffer to leverage learning across tasks in a resource-efficient manner. Although rehearsal-based methods are fairly effective in challenging CL scenarios, they suffer from overfitting on buffered samples (Bhat et al., 2022a), exacerbated representation drift at the task boundary (Jeeveswaran et al., 2023), and prior information loss (Zhang et al., 2020) in low buffer regimes, thus hurting the generalizability of the model.

**Regularization in Experience-Rehearsal:** Regularization, implicit or explicit, is an important component in reducing the generalization error in DNNs. Although the parameter norm penalty is one way to regularize the CL model, parameter sharing using multitask learning (Caruana, 1997) can lead to better generalization and lower generalization error bounds if there is a valid statistical relationship between tasks (Baxter, 1995). Contrastive representation learning (Chen et al., 2020b; He et al., 2020; Henaff, 2020) that solves pretext prediction tasks to learn generalizable representations across a multitude of downstream tasks is an ideal candidate as an auxiliary task for implicit regularization. In CL, TARC (Bhat et al., 2022b) proposes a two-stage learning paradigm in which the model learns generalizable representations first using Supervised Contrastive (SupCon) (Khosla et al., 2020) loss, followed by a modified supervised learning stage. Similarly, Co$^2$L (Cha et al., 2021) first learns representations using modified SupCon loss and then trains a classifier only on the last task samples and buffer data. However, these approaches require training in two phases and knowledge of the task boundary. OCDNet (Li et al., 2022) employs a student model and distills relational and adaptive knowledge using a modified SupCon objective. However, OCDNet does not leverage the generic information captured within the projection head to further reduce the overfitting of the classifier.

Explicit regularization in the function space imposes soft constraints on the parameters and optimizes the learning goal to converge upon a function that maps inputs to outputs.Therefore, several methods opt to directly limit how much the input/output function changes between tasks to promote generalization. Dark Experience Replay (DER++) (Buzzega et al., 2020) saves the model responses in the buffer and applies

consistency regularization while replaying data from the memory buffer. Instead of storing the responses in the buffer, the Complementary Learning System-based ER (CLS-ER) (Arani et al., 2022) maintains dual semantic memories to enforce consistency regularization. However, in addition to consistency regularization, these approaches might further benefit from multitasking and explicit classifier regularization to enable further generalization in CL.

**Inter-task class separation:**  The problem of inter-task class separation in Class-IL remains a significant challenge due to the difficulty in establishing clear boundaries between classes of current and previous tasks (Lesort et al., 2019). When a limited number of samples from previous tasks are available in the buffer, the CL model tends to overfit on the buffered samples and incorrectly approximates the class boundaries between classes from current and previous tasks. Kim & Choi (2021) splits the Class-IL problem into intra-old, intra-new and cross-task knowledge. The cross-task knowledge specifically addresses the inter-task class separation through knowledge distillation. Similarly, Kim et al. (2022) decomposes the Class-IL into two sub-problems: within-task prediction (WP) and task-id prediction (TP). Essentially, a good WP and good TP or out-of-distribution detection are necessary and sufficient for good Class-IL performance (Kim et al., 2022). On the other hand, some approaches propose to address inter-task forgetting without decomposition. Attractive and repulsive training (ART) (Choi & Choi, 2022), which effectively captures the previous feature space into a set of class-wise flags, and thereby makes old and new similar classes less correlated in the new feature space. LUCIR (Hou et al., 2019) incorporates cosine normalization, less-forget constraint, and inter-class separation, to mitigate the adverse effects of the imbalance in Class-IL. Although these approaches address fine grained problem of inter-task class separation in Class-IL, catastrophic forgetting is greatly reduced as a direct consequence.

**Memory Overfitting:**  Rehearsal-based approaches that store and replay previous task samples have been quite successful in mitigating forgetting across CL scenarios. These strategies, however, suffer from a common pitfall: as the memory buffer only stores a tiny portion of historical data, there is a significant chance that they may overfit, hurting generalization (Verwimp et al., 2021). Therefore, several methods resort to augmentation techniques either by combining multiple data points into one (Boschini et al., 2022) or by producing multiple versions of the same buffer data point (Bang et al., 2021). Gradient-based Memory EDiting (GMED) (Jin et al., 2021) proposes to create more "challenging" examples for replay by providing a framework for editing stored examples in continuous input space via gradient updates. Instead of individually editing memory data points without considering distribution uncertainty, Distributionally Robust Optimization (DRO) (Wang et al., 2022b) framework focuses on population-level and distribution level evolution. On the other hand, Lipschitz Driven Rehearsal (LiDER) (Bonicelli et al., 2022) proposes a surrogate objective that induces smoothness in the backbone network by constraining its layer-wise Lipschitz constants w.r.t. replay examples. As many of these approaches are intended to be orthogonal to the existing rehearsal-based approaches, they allow for seamless integration and effective reduction in memory overfitting in CL.

**Inductive bias :**  Fundamentally, learning benefits from prior knowledge about the types of problems to be solved. The incorporation of such prior knowledge into an optimization system is facilitated through inductive biases. In the context of the brain, continual learning (CL) is underpinned by a variety of neurophysiological processes that embody robust inductive biases, fostering the acquisition of generalizable features that demand minimal adaptation when confronted with novel tasks (Kudithipudi et al., 2022). Significantly, these inductive biases have evolved to enhance an organism's adaptability and survival within the broader ecological landscape (Richards et al., 2019). Notably, (i) many species, particularly humans, undergo protracted developmental periods characterized by extensive experiential learning, and (ii) deep neural networks have demonstrated proficiency in low-data settings owing to their effective inductive biases (Snell et al., 2017).

Traditionally, regularization techniques have served to imbibe DNNs with inductive biases, favoring certain hypotheses over others and promoting generalization. To this end, we propose intertwining implicit and explicit regularization to promote generalization in CL under low buffer regimes. This leverages generic representations learned within the projection head to compensate for weak supervision under low buffer regimes.

## 3 Method

Continual learning typically consists of $t \in \{1, 2, .., T\}$ sequence of tasks with the model learning one task at a time. Each task is specified by a task-specific data distribution $\mathcal{D}_t$ with $\{(x_i, y_i)\}_{i=1}^N$ pairs. Our CL model $\Phi_\theta = \{f, g_{lin}, g_{mlp}, h\}$ consists of four randomly initialized, learnable components: a shared backbone $f$, a linear classifier $g_{lin}$, an MLP classifier projection $g_{mlp}$, and a projection head $h$. The classifier $g_{lin}$ represents all the classes that belong to all the tasks, and the projection head $h$ captures the embeddings of the $\ell_2$-normalized representation. Classifier embeddings are further projected onto a unit hypersphere using another projection network $g_{mlp}$. CL is especially challenging when data pertaining to previous tasks vanish as the CL model progresses to the next task. Therefore, to approximate the task-specific data distributions seen previously, we seek to maintain a memory buffer $\mathcal{D}_m$ using *Reservoir sampling* (Vitter, 1985) (Algorithm 2). To restrict the empirical risk on all tasks seen so far, ER minimizes the following objective:

$$\mathcal{L}_{er} = \frac{1}{|b|} \sum_{(x,y)\sim b,\ b\in\mathcal{D}_t\cup\mathcal{D}_m} \mathcal{L}_{ce}(\sigma(g_{lin}(f(x))), y) \tag{1}$$

where $b$ is a training batch, $\mathcal{L}_{ce}$ is cross-entropy loss, $t$ is the index of the current task, and $\sigma(.)$ is the softmax function. When the buffer size is limited, the CL model learns features specific to buffered samples rather than representative features that are class- or task-wide, resulting in poor performance on previously seen tasks. Therefore, we propose IMEX-Reg, aimed at implicit regularization using parameter sharing and multitask learning, and explicit regularization in the function space to guide the optimization of the CL model towards generalization. We describe in detail the different components of our approach in the following sections.

### 3.1 Implicit regularization

We seek to learn an auxiliary task that complements continual supervised learning. We consider CRL using SupCon (Khosla et al., 2020) loss as an auxiliary task to accumulate generalizable representations in shared parameters. Ideally, CRL involves highly correlated multiple augmented views of the same sample which are then propagated forward through the encoder $f$ and the projection head $h$. It is a common practice within CRL literature to employ a non-linear projection head $h$ after CNN backbone $f$ as it improves representation quality (Chen et al., 2020c). To learn visual representations, the CL model should learn to maximize cosine similarity ($\ell_2$-normalized dot product) between the positive pairs of multiple views while simultaneously pushing away the negative embeddings from the rest of the batch. The loss takes the following form:

$$\mathcal{L}_{rep} = \sum_{i\in I} \frac{-1}{|P(i)|} \sum_{p\in P(i)} \left[\ \langle \boldsymbol{z}_i \cdot \boldsymbol{z}_p \rangle / \tau - \log \sum_{n\in N(i)} \exp\left(\langle \boldsymbol{z}_i \cdot \boldsymbol{z}_n \rangle / \tau\right)\ \right] \tag{2}$$

where $z = h(f(.))$ is any arbitrary 128-dimensional $\ell_2$-normalized projection, $\tau$ is a temperature parameter, $I$ is a set of $b$ indices, $N(i) \equiv I\backslash\{i\}$ is a set of negative indices, $P(i) \equiv \{p \in N(i) : \boldsymbol{y}_p = \boldsymbol{y}_i\}$ is a set of projection indices that belong to the same class as the anchor $z_i$ and $|P(i)|$ is its cardinality. In the following conjecture, we provide intuition behind choosing CRL as an auxiliary task in CL.

**Conjecture 1.** *(Feature similarity) Features learned by $f$ through CRL are similar to those learned via cross-entropy as long as: (i) The augmentation in CRL do not corrupt semantic information, and (ii) The labels in cross-entropy rely mainly on this semantic information (Wen & Li, 2021).*

Let $x_p^+$ and $x_p^{++}$ be two augmented positive samples such that $y_p^+ = y_p^{++}$. Furthermore, we assume that our raw data samples are generated in the following form: $x_p = \zeta_p + \xi_p$ where $\zeta_p$ represents semantic information in the image, while $\xi_p \sim \mathcal{D}_\xi = \mathcal{N}(0, \sigma)$ represents spurious noise. Given semantic preserving augmentations, (Wen & Li, 2021) state that similar discriminative features are learned by contrastive learning and cross-entropy. Similarly, since the CRL in Equation 2 employs both semantic preserving augmentations and labels to create positive pairs, we assume that the inner product of semantic information $\langle z_{\zeta_p^+}, z_{\zeta_p^{++}} \rangle$ will overwhelm that of the noisy signal $\langle z_{\xi_p^+}, z_{\xi_p^{++}} \rangle$. As we expect labels in cross-entropy to focus on semantic features to learn classification, we hypothesize that both CRL and cross-entropy share a common hypothesis class and sharing representations across these tasks especially benefits CL under low buffer regimes.

## 3.2 Explicit regularization

A CL model equipped with multitask learning implicitly encourages the shared encoder $f$ to learn generalizable features. However, the classifier $g_{lin}$ that decides the final predictions is still prone to overfitting on buffered samples under low buffer regimes. Therefore, we seek to explicitly regularize the CL model in the function space defined by the classifier $g_{lin}$. To this end, we denote the output activation of the encoder $f$ as $F \in \mathbb{R}^{b \times D_f}$, that of the projection head $h$ as $Z \in \mathbb{R}^{b \times D_h}$, and that of the classifier projection $g_{mlp}$ as $C \in \mathbb{R}^{b \times D_g}$, where $D_f, D_g, D_h$ denote the dimensions of output Euclidean spaces. Let $\mathcal{F}_g : \mathbb{R}^{D_f} \to \mathbb{R}^{D_g}$ and $\mathcal{F}_h : \mathbb{R}^{D_f} \to \mathbb{R}^{D_h}$ be the function spaces represented by the classifier $g_{lin}$ and the projection head $h$. Let $\theta$ and $\theta_{EMA}$ be parameters of the CL model and its corresponding exponential moving average (EMA). Following CLS-ER (Arani et al., 2022), we stochastically update the EMA model as follows:

$$\theta_{EMA} = \begin{cases} \eta\, \theta_{EMA} + (1 - \eta)\, \theta, & \text{if } \gamma \geq \mathcal{U}(0, 1) \\ \theta_{EMA}, & \text{otherwise} \end{cases} \tag{3}$$

where $\eta$ is a decay parameter and $\gamma$ is an update rate. The EMA of a model can be considered to form a self-ensemble of intermediate model states that leads to a better internal representation (Arani et al., 2022). Therefore, we leverage the soft targets (predictions) of the EMA model to regularize the learning trajectory in the function spaces $\mathcal{F}_g$ and $\mathcal{F}_h$ of the CL model:

$$\begin{aligned} \mathcal{L}_{cr}^g &\triangleq \mathop{\mathbb{E}}_{(x_j, y_j) \sim \mathcal{D}_m} \|\hat{y} - \hat{y}_e\|_F^2 \\ \mathcal{L}_{cr}^h &\triangleq \mathop{\mathbb{E}}_{(x_j, y_j) \sim \mathcal{D}_m} \|z - z_e\|_F^2 \end{aligned} \tag{4}$$

where $\|\cdot\|_F$ is the Frobenius norm, $z$ and $\hat{y}$ are the projection head and classifier responses of the CL model, respectively, and $z_e$ and $\hat{y}_e$ are those of the EMA model. As soft targets carry more information per training sample than ground truth labels (Hinton et al., 2015), knowledge of previous tasks can be better preserved by ensuring consistency in predictions, leading to drastic reductions in overfitting.

It is pertinent to note that restricting the output space to a unit hypersphere in representation learning not only enhances training stability but also creates well-clustered projections in the hypersphere that are linearly separable from the rest of the samples (Wang & Isola, 2020). We hypothesize that by regularizing the classifier's unit hypersphere using the projection head's unit hypersphere could potentially guide the classifier's decision space to discern more effective boundaries. As semantically similar inputs tend to elicit similar responses, we seek to align geometric structures within the classifier's hypersphere with that of the projection head's hypersphere to further leverage the global relationship between samples established using instance discrimination task. We assume that there exist a mapping function $\mathcal{M} : \mathbb{R}^{D_h} \to \mathbb{R}^{D_g}$ and its inverse $\mathcal{M}^{-1} : \mathbb{R}^{D_g} \to \mathbb{R}^{D_h}$ that establish a connection between the geometric relationship between the points in both hyperspheres. When learning in the hypersphere, angular information rather than magnitude forms the key semantics in the CL model (Chen et al., 2020a). Therefore, to guide the classifier toward the activation correlations in the unit hypersphere of the projection head, we regularize the differences in the outer products of $Z$ and $C$, i.e.,

$$\begin{aligned} G_h &= ZZ^T \in \mathbb{R}^{b,b} \\ G_g &= CC^T \in \mathbb{R}^{b,b} \end{aligned} \tag{5}$$

$$\mathcal{L}_{ecr} = \frac{1}{|b|^2} \left\| stopgrad(G_h) - G_g \right\|_F^2 \tag{6}$$

where $stopgrad(.)$ ensures that the backpropagation of gradients occurs only through the classifier. Equation 4 regularizes both the classifier and the projection head using the EMA of the CL model, while $\mathcal{L}_{ecr}$ in Equation 6 captures the mean element-wise squared difference between $G_h$ and $G_g$ matrices of the CL model.

**Theorem 2.** *(Johnson-Lindenstrauss Lemma): Let $\epsilon \in (0, 1)$ and $D_g > 0$ be such that for any integer $n$, $D_g \geq 4\left(\epsilon^2/2 - \epsilon^3/3\right)^{-1} \ln n$. Then, for any set of points $Z \in \mathbb{R}^{D_h}$, there exists a mapping function $\mathcal{M} : \mathbb{R}^{D_h} \to \mathbb{R}^{D_g}$ such that for all pairs of samples $p, q$*

$$(1 - \epsilon)\|p - q\|^2 \leq \|\mathcal{M}(p) - \mathcal{M}(q)\|^2 \leq (1 + \epsilon)\|p - q\|^2. \tag{7}$$

---

**Algorithm 1** Proposed Method: IMEX-Reg

---

1: **Input:** Data streams $\mathcal{D}_t$, Model $\Phi_\theta = \{f, g, g_{mlp}, h\}$, Hyperparameters $\alpha$, $\beta$ and $\lambda$, Memory buffer $\mathcal{D}_m \leftarrow \{\}$
2: **for all** tasks $t \in \{1, 2, .., T\}$ **do**
3:      **for all** Iterations $e \in \{1, 2, .., E\}$ **do**
4:          $\mathcal{L} = 0$
5:          Sample a minibatch $(X_t, Y_t) \in \mathcal{D}_t$
6:          $F_t = f(X_t)$
7:          $\hat{Y}_t, Z_t, C_t = g(F_t), h(F_t), g_{mlp}(g(F_t))$
8:          **if** $\mathcal{D}_m \neq \emptyset$ **then**
9:              Sample a minibatch $(X_m, Y_m) \in \mathcal{D}_m$
10:             $F_m = f(X_m)$
11:             $\hat{Y}, Z, C = g(F_m), h(F_m), g_{mlp}(g(F_m))$
12:             $F_e = f_e(X_m)$
13:             $\hat{Y}_e, Z_e, C_e = g_e(F_m), h_e(F_m), g_{mlp_e}(g_e(F_m))$
14:             $\mathcal{L} \mathrel{+}= \lambda \left[ \mathcal{L}_{cr}^g + \mathcal{L}_{cr}^h \right]$                          $\triangleright$ Equation 4
15:          $\mathcal{L} \mathrel{+}= \mathcal{L}_{er} + \alpha\, \mathcal{L}_{rep} + \beta\, \mathcal{L}_{ecr}$            $\triangleright$ Equations 1, 2, and 6
16:          Update $\Phi_\theta$ and $\mathcal{D}_m$
17:          Update $\theta_{ema}$                                    $\triangleright$ Equation 3
18: **return** model $\Phi_\theta$

---

Johnson-Lindenstrauss (JL) lemma (Dasgupta & Gupta, 2003) states that any $p$ points in a high dimensional Euclidean space can be mapped onto $k$ dimensions where $k \geq O\left(\log p / \epsilon^2\right)$ without distorting the Euclidean distance between any two points more than a factor of $1 \pm \epsilon$. Under JL lemma, we hypothesize that it is possible to map geometric relationship between points in a higher dimensional projection head hypersphere to a lower dimensional classifier hypersphere without the loss of generality. To this end, we propose a mapping function $\mathcal{L}_{ecr}$ that preserves the geometric structures when mapping from projection head to classifier. Application of JL lemma in this context implies that the distortion in geometric relationship between points when mapping from projection head to classifier hypersphere can be limited to $1 \pm \epsilon$.

### 3.3 Putting it all together

During training, the batches of the current task are propagated forward through $\Phi_\theta$ to obtain classification and projection embeddings. Specifically, $\Phi_\theta$ learns generalizable features through Equation 2 and task-specific features through Equation 1. To better consolidate the information pertaining to previous tasks, we maintain a memory buffer $\mathcal{D}_m$ and an EMA of the CL model, which also serves as an inference model for evaluation. We enforce consistency in predictions on rehearsal data using Equation 4. To further reduce overfitting and discourage label bias in the classifier, we seek to emulate geometric structures using Equation 6. During each training iteration, the memory buffer is updated using Reservoir sampling (Vitter, 1985) and the EMA is stochastically updated using Equation 3. The overall learning objective is as follows:

$$\mathcal{L} \triangleq \mathop{\mathbb{E}}_{(x,y)\sim\mathcal{D}_t\cup\mathcal{D}_m} \left[ \mathcal{L}_{er} + \alpha\, \mathcal{L}_{rep} + \beta\, \mathcal{L}_{ecr} \right] + \mathop{\mathbb{E}}_{(x,y)\sim\mathcal{D}_m} \lambda \left[ \mathcal{L}_{cr}^g + \mathcal{L}_{cr}^h \right] \tag{8}$$

where $\alpha$, $\beta$ and $\lambda$ are hyperparameters. Except for *stopgrad*(.) in Equation 5, the entire network receives weight updates through different loss functions in Equation 8. IMEX-Reg is illustrated in Figure 1 and is detailed in Algorithm 1.

## 4 Experiments

### 4.1 Experimental setup

We build on top of the Mammoth (Buzzega et al., 2020) CL repository in PyTorch. We evaluate CL models under Class-Incremental Learning (Class-IL), Task-Incremental Learning (Task-IL), and Generalized Class-IL

Table 1: Top-1 accuracy (%) of different CL models in Class-IL and Task-IL scenarios with varying complexities and memory buffer sizes. The best results are marked in bold.

| Buffer | Methods | Venue | Seq-CIFAR10 | | Seq-CIFAR100 | | Seq-TinyImageNet | |
|---|---|---|---|---|---|---|---|---|
| | | | Class-IL | Task-IL | Class-IL | Task-IL | Class-IL | Task-IL |
| - | SGD | - | $19.62_{\pm0.05}$ | $61.02_{\pm3.33}$ | $17.49_{\pm0.28}$ | $40.46_{\pm0.99}$ | $07.92_{\pm0.26}$ | $18.31_{\pm0.68}$ |
| | Joint | - | $92.20_{\pm0.15}$ | $98.31_{\pm0.12}$ | $70.56_{\pm0.28}$ | $86.19_{\pm0.43}$ | $59.99_{\pm0.19}$ | $82.04_{\pm0.10}$ |
| 200 | ER | - | $44.79_{\pm1.86}$ | $91.19_{\pm0.94}$ | $21.40_{\pm0.22}$ | $61.36_{\pm0.35}$ | $8.57_{\pm0.04}$ | $38.17_{\pm2.00}$ |
| | ER-ACE | ICLR'22 | $62.08_{\pm1.44}$ | $92.20_{\pm0.57}$ | $35.17_{\pm1.17}$ | $63.09_{\pm1.23}$ | $11.25_{\pm0.54}$ | $44.17_{\pm1.02}$ |
| | GCR | CVPR'22 | $64.84_{\pm1.63}$ | $90.8_{\pm1.05}$ | $33.69_{\pm1.40}$ | $64.24_{\pm0.83}$ | $13.05_{\pm0.91}$ | $42.11_{\pm1.01}$ |
| | DRI | AAAI'22 | $65.16_{\pm1.13}$ | $92.87_{\pm0.71}$ | - | - | $17.58_{\pm1.24}$ | $44.28_{\pm1.37}$ |
| | DER++ | NeurIPS'20 | $64.88_{\pm1.17}$ | $91.92_{\pm0.60}$ | $29.60_{\pm1.14}$ | $62.49_{\pm1.02}$ | $10.96_{\pm1.17}$ | $40.87_{\pm1.16}$ |
| | CLS-ER | ICLR'22 | $66.19_{\pm0.75}$ | $93.90_{\pm0.60}$ | $43.80_{\pm1.89}$ | $73.49_{\pm1.04}$ | $23.47_{\pm0.80}$ | $49.60_{\pm0.72}$ |
| | $Co^2L$ | ICCV'21 | $65.57_{\pm1.37}$ | $93.43_{\pm0.78}$ | $31.90_{\pm0.38}$ | $55.02_{\pm0.36}$ | $13.88_{\pm0.40}$ | $42.37_{\pm0.74}$ |
| | OCDNet | IJCAI'22 | $\mathbf{73.38}_{\pm0.32}$ | $\mathbf{95.43}_{\pm0.30}$ | $44.29_{\pm0.49}$ | $73.53_{\pm0.24}$ | $17.60_{\pm0.97}$ | $56.19_{\pm1.31}$ |
| | IMEX-Reg | - | $71.56_{\pm0.18}$ | $94.77_{\pm0.81}$ | $\mathbf{48.54}_{\pm0.23}$ | $\mathbf{75.61}_{\pm0.73}$ | $\mathbf{24.15}_{\pm0.78}$ | $\mathbf{62.91}_{\pm0.54}$ |
| 500 | ER | - | $57.74_{\pm0.27}$ | $93.61_{\pm0.27}$ | $28.02_{\pm0.31}$ | $68.23_{\pm0.17}$ | $9.99_{\pm0.29}$ | $48.64_{\pm0.46}$ |
| | ER-ACE | ICLR'22 | $68.45_{\pm1.78}$ | $93.47_{\pm1.00}$ | $40.67_{\pm0.06}$ | $66.45_{\pm0.71}$ | $17.73_{\pm0.56}$ | $49.99_{\pm1.51}$ |
| | GCR | CVPR'22 | $74.69_{\pm0.85}$ | $94.44_{\pm0.32}$ | $45.91_{\pm1.30}$ | $71.64_{\pm2.10}$ | $19.66_{\pm0.68}$ | $52.99_{\pm0.89}$ |
| | DRI | AAAI'22 | $72.78_{\pm1.44}$ | $93.85_{\pm0.46}$ | - | - | $22.63_{\pm0.81}$ | $52.89_{\pm0.60}$ |
| | DER++ | NeurIPS'20 | $72.70_{\pm1.36}$ | $93.88_{\pm0.50}$ | $41.40_{\pm0.96}$ | $70.61_{\pm0.08}$ | $19.38_{\pm1.41}$ | $51.91_{\pm0.68}$ |
| | CLS-ER | ICLR'22 | $75.22_{\pm0.71}$ | $94.94_{\pm0.53}$ | $51.40_{\pm1.00}$ | $78.12_{\pm0.24}$ | $31.03_{\pm0.56}$ | $60.41_{\pm0.50}$ |
| | $Co^2L$ | ICCV'21 | $74.26_{\pm0.77}$ | $95.90_{\pm0.26}$ | $39.21_{\pm0.39}$ | $62.98_{\pm0.58}$ | $20.12_{\pm0.42}$ | $53.04_{\pm0.69}$ |
| | OCDNet | IJCAI'22 | $\mathbf{80.64}_{\pm0.77}$ | $\mathbf{96.57}_{\pm0.07}$ | $54.13_{\pm0.36}$ | $78.51_{\pm0.24}$ | $26.09_{\pm0.28}$ | $64.76_{\pm0.29}$ |
| | IMEX-Reg | - | $77.61_{\pm0.18}$ | $95.96_{\pm0.33}$ | $\mathbf{56.53}_{\pm0.80}$ | $\mathbf{80.51}_{\pm0.10}$ | $\mathbf{31.41}_{\pm0.21}$ | $\mathbf{67.44}_{\pm0.38}$ |

(GCIL) (Van de Ven & Tolias, 2019; Arani et al., 2022). More information on the datasets, task partition, and the corresponding network architecture used in these scenarios can be found in Appendix A. To provide a comprehensive analysis, we compare IMEX-Reg with several approaches that aim to improve generalization under low buffer regimes in CL. We consider ER-ACE, GCR, DRI (aimed at improving generalization), DER++, CLS-ER (use explicit consistency regularization by leveraging soft targets), $Co^2L$ and OCDNet (representation and/or auxiliary CRL) as our baselines. Furthermore, we provide a lower bound 'SGD', without using any mechanism to minimize catastrophic forgetting, and an upper bound 'Joint', where the training is carried out using the entire dataset. We report the average accuracy along with the standard deviation on all tasks after CL training with three random seeds. We also provide the results of the forgetting analysis.

## 4.2 Experimental results

Table 1 presents the comparison of our method with the baselines for the Class-IL and Task-IL settings. Several observations can be made from these results: (i) Although CL methods aimed at generalization improve over ER, they lack strong inductive biases proposed in this work, thus failing to make significant improvements in low-buffer regimes. (ii) Explicit consistency regularization shows great promise in reducing overfitting over ER. As can be seen, DER++, DRI, and CLS-ER show significant reductions in catastrophic forgetting compared to ER. However, IMEX-Reg incorporates implicit and explicit regularization to promote generalization and outperforms these methods by a large margin in most scenarios. In Seq-TinyImageNet, IMEX-Reg outperforms CLS-ER by a relative margin of 2.9% and 1.2% in buffer sizes 200 and 500, respectively. Note that CLS-ER employs two semantic memories, while IMEX-Reg uses only one.

$Co^2L$ and OCDNet generalize well across tasks in CL, showing the efficacy of CRL in CL. Specifically, OCDNet combines consistency regularization and CRL in addition to self-supervised rotation prediction. OCDNet outperforms IMEX-Reg in Seq-CIFAR10 in both buffer sizes, greatly benefiting from rotation prediction. However, OCDNet lags behind IMEX-Reg by a large margin in more challenging datasets. Essentially, IMEX-Reg compensates the classifier's weak supervision with the generic geometric structures learned by the

Table 2: Top-1 accuracy (%) of different CL models for Uniform and Longtail GCIL-CIFAR100 settings with different memory buffer sizes. The best results are marked in bold.

| Method | Uniform | | | Longtail | | |
|---|---|---|---|---|---|---|
| SGD | 10.38±0.26 | | | 9.61±0.19 | | |
| Joint | 58.59±1.95 | | | 58.42±1.32 | | |
| Buffer | 100 | 200 | 500 | 100 | 200 | 500 |
| ER | 14.43±0.36 | 16.52±0.10 | 23.62±0.66 | 13.22±0.6 | 16.20±0.30 | 22.36±1.27 |
| ER-ACE | 23.76±1.61 | 27.64±0.76 | 30.14±1.11 | 23.05±0.18 | 25.10±2.64 | 31.88±0.73 |
| DER++ | 21.17±1.65 | 27.73±0.93 | 35.83±0.62 | 20.29±1.03 | 26.48±2.07 | 34.23±1.19 |
| CLS-ER | 33.42±0.30 | 35.88±0.41 | 38.94±0.38 | 33.92±0.79 | 35.67±0.72 | 38.79±0.67 |
| OCDNet | 37.21±0.69 | 39.94±0.05 | 43.58±0.67 | 35.61±1.13 | 39.97±0.70 | 43.57±0.23 |
| IMEX-Reg | **39.48**±0.25 | **43.19**±0.47 | **49.07**±0.59 | **38.39**±0.46 | **42.66**±0.82 | **46.81**±1.04 |

Table 3: Forgetting analysis for various CL models across datasets in Class-IL setting.

| Buffer | Methods | Seq-CIFAR10 | Seq-CIFAR100 | Seq-TinyImageNet |
|---|---|---|---|---|
| 200 | ER | 61.24±2.62 | 75.54±0.45 | 76.37±0.53 |
| | DER++ | 32.59±2.32 | 68.77±1.72 | 72.74±0.56 |
| | OCDNet | **22.63**±2.06 | 33.41±2.87 | **20.37**±1.05 |
| | IMEX-Reg | 24.69±0.84 | **32.19**±0.51 | 27.93±2.99 |
| 500 | ER | 45.35±0.07 | 67.74±1.29 | 75.27±0.17 |
| | DER++ | 22.38±4.41 | 50.99±2.52 | 64.58±2.01 |
| | OCDNet | **14.93**±1.42 | 21.6±0.16 | **17.88**±0.55 |
| | IMEX-Reg | 17.00±0.40 | **18.83**±0.51 | 30.47±0.35 |

projection head through an explicit regularization, achieving a relative 9.6% and 37.22% improvement over OCDNet in Seq-CIFAR100 and Seq-TinyImageNet, respectively, for a low buffer size of 200. This reinforces our earlier hypothesis in Conjecture 1 that leveraging desirable traits from CRL can indeed guide the classifier towards generalization under low buffer regimes.

Generalized Class-IL (GCIL) exposes the CL model to a more challenging and realistic learning scenario by using probabilistic distributions to sample data from the CIFAR100 dataset in each task (Mi et al., 2020). Table 2 shows the comparison of different CL methods in GCIL setting under two variations, Uniform and Longtail (class imbalance). As can be seen, IMEX-Reg outperforms all baselines by a large margin across all buffer sizes. IMEX-Reg performs significantly better than ER, ER-ACE, DER++ and CLS-ER, emphasizing the importance of implicit regularization through auxiliary CRL in improving generalization. Although OCDNet combines auxiliary CRL and explicit consistency regularization, OCDNet falls behind IMEX-Reg by a considerable margin across all GCIL scenarios. Even under class imbalance (Longtail) and a low buffer size of 100, IMEX-Reg achieves a relative improvement of 7.8% over OCDNet. These results indicate that IMEX-Reg learns generalizable features through auxiliary CRL while enriching the classifier through explicit regularization in the function space, making it suitable for challenging CL scenarios.

## 4.3 Forgetting Analysis

Continually learning on a sequence of novel tasks often interferes with previously learned information resulting in catastrophic forgetting. Therefore, in addition to the average accuracy of the previous tasks, it is crucial to measure how much of the learned information is preserved. The forgetting measure ($f_j^k$) (Chaudhry et al., 2018) for a task $j$ after learning $k$ tasks is defined as the difference between the maximum knowledge gained for the task in the past and the knowledge currently in the model. Let $A_{ij}$ be the test accuracy of the model for task $j$ after learning task $i$ then,

$$f_j^k = \max_{l \in \{j, j+1, .., k-1\}} A_{lj} - A_{kj}, \forall j < k \quad (9)$$

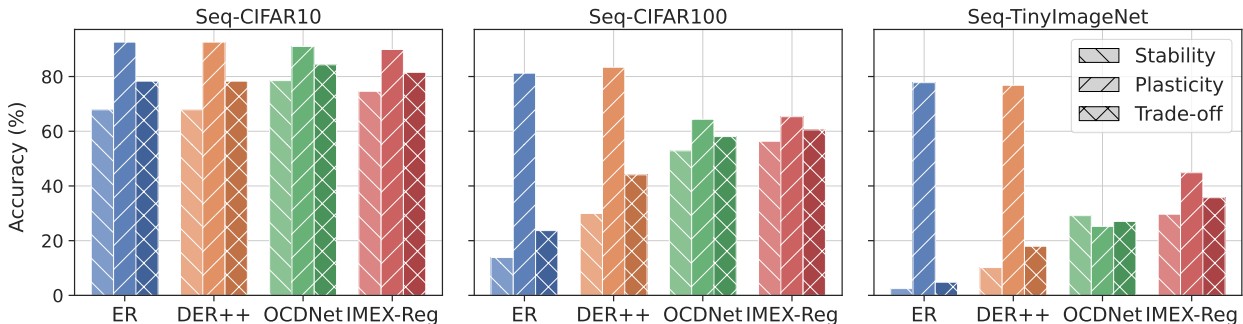

Figure 2: Comparison of Stability-Plasticity Trade-off for different CL models across different datasets.

Then the average forgetting measure for the model after learning $T$ tasks is given by

$$F_T = \frac{1}{T-1} \sum_{j=0}^{T-1} f_j^T \tag{10}$$

The lower $F_T$ signifies less forgetting of previous tasks. Typically, $A_{ij}$ is computed at the task boundary after learning the task $i$. However, since the EMA is updated stochastically, the maximum accuracy for a previous task is not necessarily achieved at the task boundaries. Therefore, we evaluate the EMA on previous tasks after every epoch and keep track of the maximum accuracy observed for the previous tasks. Forgetting analysis of IMEX-Reg is then computed with respect to the maximum accuracy observed for the previous tasks and the final evaluation accuracy for the tasks at the end of training.

Table 3 shows the average forgetting measures for different CL methods across different datasets and buffer sizes in a Class-IL setting. IMEX-Reg and OCDNet achieve significantly lower forgetting than other baselines, owing to the ability of the EMA to preserve the consolidated knowledge. In Seq-TinyImageNet, OCDNet suffers from far less forgetting than IMEX-Reg. However, this can be attributed to how often the EMA is updated. By restricting the update frequency of EMA, we can considerably reduce forgetting but at the cost of adapting to new information. An ideal CL model should not be too restrictive and instead should find an optimal balance between forgetting and learning novel tasks.

## 4.4 Stability-Plasticity Trade-off

Figure 2 compares the stability, plasticity, and trade-off (Appendix A.10) of different CL methods across different datasets for a buffer size of 500. ER and DER++ have high plasticity and quickly adapt to novel information; yet, they do not retain previously learned information. On the other hand, IMEX-Reg and OCDNet employ an EMA that is stochastically updated and better preserves consolidated knowledge. Hence, both IMEX-Reg and OCDNet achieve much higher stability at a relatively lower cost of plasticity, resulting in a higher stability-plasticity trade-off. In Table 3, we see that OCDNet suffered the least forgetting in Seq-TinyImageNet. However, Figure 2 shows that OCDNet achieved the least plasticity in Seq-TinyImageNet. IMEX-Reg, on the other hand, achieved a much better trade-off, even though it forgets more than OCDNet.

## 5 Model Characteristics

**Task Recency Bias.** Learning continuously on a sequence of tasks biases the model predictions toward recently learned tasks in the Class-IL scenario (Hou et al., 2019). An ideal CL model is expected to have the least bias, with predictions evenly distributed across all tasks. To gauge the task-recency bias in IMEX-Reg, we compute the average task probabilities by averaging the softmax outputs of all samples associated with each task on the test set at the end of the training. Figure 3(right) shows the task-recency bias of different CL models trained on Seq-CIFAR100 with buffer size 200. Evidently, IMEX-Reg predictions are more evenly

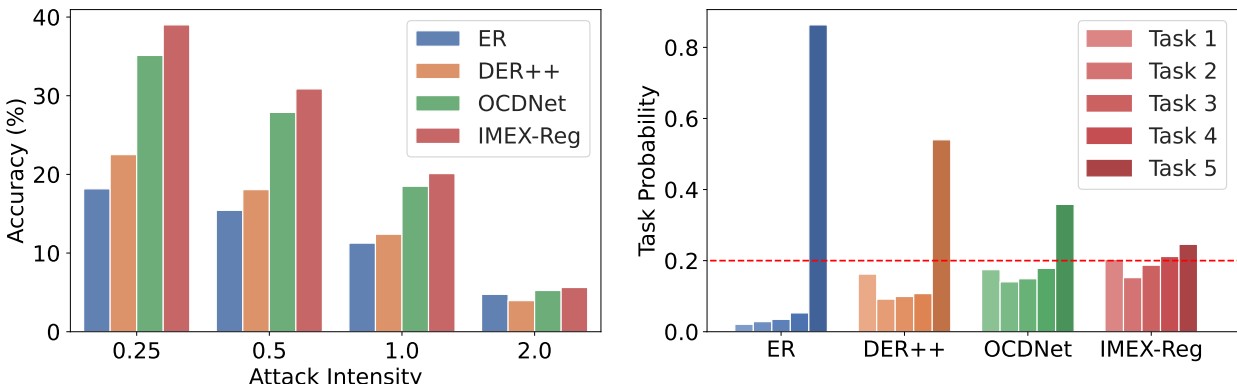

Figure 3: (Left) Robustness to PGD adversarial attack at varying strengths and (Right) Average probability of predicting each task for different CL methods trained on Seq-CIFAR100 with 5 tasks. IMEX-Reg shows the highest robustness and the least recency bias with probabilities evenly distributed across tasks.

distributed with least recency bias. IMEX-Reg induces robust inductive biases that skew the learning process towards generic representations instead of aligning more with the current task, thereby reducing recency bias.

**Robustness to Adversarial Attacks.** Adversarial attacks generate specially crafted images with imperceptible perturbations to fool the network into making false predictions (Szegedy et al., 2013). We analyze adversarial robustness by performing a PGD-10 attack (Madry et al., 2017) with varying attack intensities on different models trained on Seq-CIFAR100 with a buffer size of 200. Figure 3(left) shows that IMEX-Reg is more resistant to adversarial attacks compared to other baselines. OCDNet and IMEX-Reg are significantly more robust than DER++ and ER even at higher attack intensities, owing to the auxiliary CRL that encourages the model to learn generalizable features. However, IMEX-Reg further biases the model towards generalization through explicit classifier regularization and outperforms OCDNet across all attack intensities. Thus, inducing right-inductive biases in the model can improve its robustness in addition to improved performance.

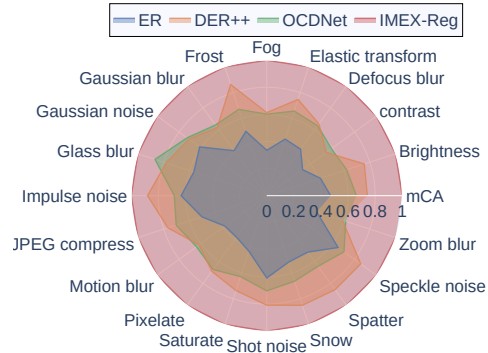

Figure 4: Relative top-1 accuracy (%) (averaged over 5 severity levels) for 19 different natural corruptions for different CL models trained on Seq-CIFAR100 with 5 tasks. The average accuracy across all corruptions is shown as mCA.

**Robustness to Natural Corruptions:** Data in the wild is often corrupted by changes in illumination, digital imaging artifacts, or weather conditions. Therefore, autonomous agents deployed in the real world should be robust to these natural corruptions, especially in safety-critical applications. To evaluate the robustness to natural corruptions, we test several CL methods trained on Seq-CIFAR00 with buffer size 500 on CIFAR100-C (Hendrycks & Dietterich, 2019). Figure 4 shows the accuracy of different CL models compared to IMEX-Reg for 19 different corruptions averaged at five severity levels. Although OCDNet combines knowledge distillation and CRL, it is unable to improve its robustness over DER++. However, IMEX-Reg further enriches the classifier with the generic geometric structures learned in the projection head, proving to be more robust to natural corruptions. Thus, the intertwining of implicit and explicit regularization in IMEX-Reg enables generalization not just across tasks, but also across domains that are drastically different from the training set.

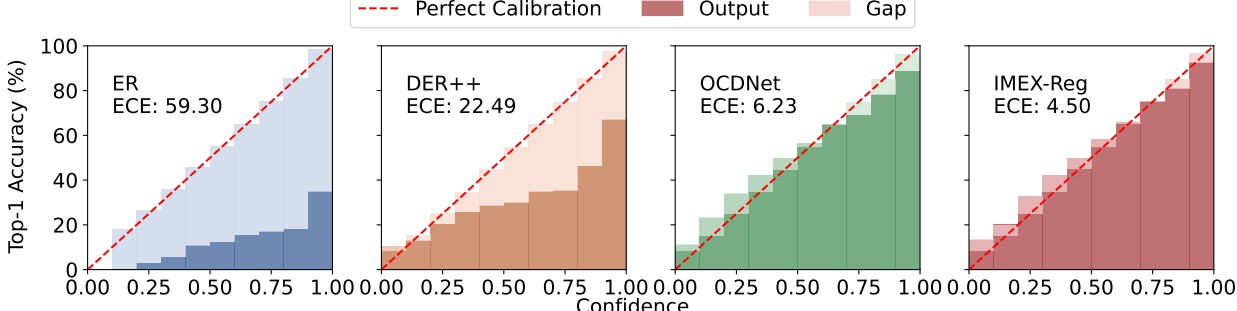

Figure 5: Reliability diagrams with Expected Calibration Error (ECE) for CL methods trained on Seq-CIFAR100 with 5 tasks. The lower ECE value signifies a better calibrated model. Compared to baselines, IMEX-Reg is well-calibrated with the lowest ECE value.

**Model Calibration** CL systems deployed in the real world are expected to be reliable by exhibiting a sufficient level of prediction uncertainty. Expected Calibration Error (ECE) provides a good estimate of reliability by gauging the difference in expectation between confidence and accuracy (Guo et al., 2017). Figure 5 shows the comparison of our method with other baselines using a calibration framework (Küppers et al., 2020). Compared to other baselines, IMEX-Reg achieves the lowest ECE value and is considerably well-calibrated. In addition to improving generalization, regularizing the classifier implicitly through the auxiliary CRL and explicitly in the function space by aligning with geometric structures learned in the projection head prevents the model from being overconfident.

## 5.1 Ablation Study

We seek to provide a better understanding of the contributions of each of the components in our proposed method. Table 4 shows the ablation study of IMEX-Reg trained on Seq-CIFAR100 for a buffer size of 200 with 5 tasks. In line with our earlier hypothesis, intertwining implicit regularization using parameter sharing and multitask learning, and explicit regularization in the function space guide the optimization of the IMEX-Reg towards better generalization representations. As can be seen, each of these components has a significant impact on the performance of IMEX-Reg under a low buffer regime. Additionally, leveraging generic geometric structures learned within CRL through explicit classifier regularization complements weak supervision under low buffer regimes and results in lower generalization error.

Table 4: Comparison of the contributions of each of the components in IMEX-Reg. The absence of $EMA$ implies consistency regularization by storing past logits.

| $\mathcal{L}_{cr}$ | $EMA$ | $\mathcal{L}_{rep}$ | $\mathcal{L}_{ecr}$ | Accuracy |
|:---:|:---:|:---:|:---:|:---|
| ✓ | ✓ | ✓ | ✓ | **48.54**±0.23 |
| ✓ | ✓ | ✓ | ✗ | 46.85±0.48 |
| ✓ | ✓ | ✗ | ✗ | 43.38±1.06 |
| ✓ | ✗ | ✗ | ✗ | 29.60±1.14 |
| ✗ | ✗ | ✗ | ✗ | 21.40±0.22 |

## 6 Conclusion

We propose IMEX-Reg, a two-pronged CL approach aimed at implicit regularization using hard parameter sharing and multitask learning, and an explicit regularization in the function space to guide the optimization of the CL model towards generalization. The novelty of our method mainly lies in emulating rich geometric structures learned within the projection head hypersphere to compensate for weak supervision under low buffer regimes. Through extensive experimental evaluation, we show that IMEX-Reg significantly benefits from each of these strong inductive biases and exhibits strong performance across several CL scenarios. Furthermore, we show that IMEX-Reg is robust and mitigates task recency bias. Leveraging unlabeled data through CRL in IMEX-Reg could further improve generalization performance across tasks in CL.

## Acknowledgments

The research was conducted when all the authors were also affiliated with Advanced Research Lab, NavInfo Europe, The Netherlands.

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

# A Appendix

## A.1 Limitations and Future Work

IMEX-Reg is a two-pronged CL approach aimed at implicit regularization using hard parameter sharing and multitask learning, and an explicit regularization in the function space to guide the optimization of the CL model toward generalization. In addition to CRL and consistency regularization, IMEX-Reg entails an explicit classifier regularization to emulate rich geometric structures learned within the projection head hypersphere to compensate for weak supervision under low-buffer regimes. As IMEX-Reg involves many inductive biases to help improve the generalization performance in CL, it necessitates hyperparameter tuning to find the best combination of these biases. Furthermore, finding the right set of inductive biases for novel domains may not be trivial. In addition, the selected inductive biases should be such that they complement each other and especially benefit CL.

As IMEX-Reg involves an auxiliary CRL, representation learning in the projection head could be further improved by leveraging the vast unlabeled data. Learning generic representations through unlabeled data could relax the need to store a large number of images in the buffer. We leave the extension of IMEX-Reg to other domains, and different CL scenarios, and enhancing with unlabeled data for future work.

## A.2 Continual Learning Settings

We evaluate the performance of our method in three different CL settings, namely, Task Incremental Learning (Task-IL), Class Incremental Learning (Class-IL), and General Continual Learning (GCL).

In Task-IL and Class-IL, each task consists of a fixed number of novel classes that the model must learn. A CL model learns several tasks sequentially while distinguishing all classes it has seen so far. Task-IL is very similar to Class-IL, with the exception that task labels are also available during inference, making it the easiest scenario. Although Class-IL is a widely studied and relatively harder CL setting, it makes several assumptions that are realistic, such as a fixed number of classes in each task and no reappearance of classes. (Mi et al., 2020). GCL relaxes such assumptions and presents more challenging real-world-like scenarios where the task boundaries are blurry and classes reappear with different distributions.

## A.3 Datasets

For Task-IL and Class-IL scenarios, we obtain Seq-CIFAR10, Seq-CIFAR100, and Seq-TinyImageNet by splitting CIFAR10, CIFAR100, and TinyImageNet into 5, 5, and 10 tasks of 2, 20, and 20 classes, respectively. The three datasets present progressively challenging scenarios (increasing the number of tasks or number of classes per task) for a comprehensive analysis of different CL methods. Generalized Class-IL (GCIL) (Mi et al., 2020) exposes the model to a more challenging scenario by utilizing probabilistic distributions to sample data from the CIFAR100 dataset in each task. The CIFAR100 dataset is split into 20 tasks, with each task containing 1000 samples with a maximum of 50 classes. GCIL provides two variations for sample distribution, Uniform and Longtail (class imbalance). GCIL is the most realistic scenario with varying numbers of classes per task and classes reappearing with different sample sizes.

## A.4 Model and Training

We use the same backbone as recent approaches in CL (Buzzega et al., 2020; Arani et al., 2022), i.e., a ResNet-18 backbone without pretraining for all experiments. We use a linear layer, 3-layer MLP with BatchNorm and ReLu, and 2-layer MLP for the classifier, projection head, and classifier projection, respectively.

To ensure uniform experimental settings, we extended the Mammoth framework (Buzzega et al., 2020) and followed the same training scheme such as the SGD optimizer, batch size, number of training epochs, and learning rate for all experiments, unless otherwise specified. We employ random horizontal flip and random crop augmentations for supervised learning in Seq-CIFAR10, Seq-CIFAR100, Seq-TinyImageNet, and GCIL-CIFAR100 experiments. For contrastive representation learning in the projection head, we transform the input batch using a stochastic augmentation module consisting of random resized crop, random horizontal

Table 5: Relative training time comparison of several CL methods trained on Seq-CIFAR100 with buffer size 200. IMEX-Reg and OCDNet takes more computational time owing to CRL

| Method | DER++ | CLS-ER | OCD-Net | IMEX-Reg |
|---|---|---|---|---|
| Relative time taken | 1x | 1.09x | 1.44x | 1.57x |

flip followed by random color distortions. We trained all our models on NVIDIA's GeForce RTX 2080 Ti (11GB). On average, it took around 2 hours to train IMEX-Reg on Seq-CIFAR10 and Seq-CIFAR100, and approximately 8 hours to train on Seq-TinyImageNet.

### A.5 Computational Cost

Table 5 provides a relative comparison of training times for different CL approaches trained on Seq-CIFAR100 with a buffer size of 200. A large part of IMEX-Reg's as well as OCDNet's training time can be attributed to computations involving CRL. Our contribution $\mathcal{L}_{ecr}$ adds up only a minimal computational overhead. Although IMEX-Reg takes longer time to train, the performance improvement from these components is significant enough to sidestep computational overhead. Moreover, during inference we discard the projection head and hence IMEX-Reg's processing time is the same as the other approaches

### A.6 Buffer Efficiency

In our experimental evaluation, we assessed various CL models using established benchmarks, encompassing a range of scenarios where the number of samples per class varies from 50 to 1. We observed that as the number of samples per class decreases, the generalization of methods tends to deteriorate, implying less effective utilization of buffer samples. However, IMEX-Reg consistently demonstrates superior performance even with a reduced number of buffer samples. For instance, we can compare the performance of DER++ with a buffer size of 500 to IMEX-Reg with a buffer size of 200. As can be seen from Table 1 and Table 2, IMEX-Reg notably outperforms DER++ in nearly all scenarios, despite having half the buffer size. This enhancement in performance can be attributed to IMEX-Reg's ability to learn generalizable representations, thus effectively leveraging its buffer samples.

### A.7 Reservoir Sampling

Algorithm 2 provides the steps for the reservoir sampling strategy (Vitter, 1985) for maintaining a fixed-size memory buffer from a data stream. Each sample in the data stream is assigned an equal probability of being represented in the memory buffer. When the buffer is full, sampling and replacement are performed at random without assigning any priority to the samples that are added or replaced.

---

**Algorithm 2** Reservoir sampling (Vitter, 1985)

---

**Input:** Data streams $\mathcal{D}_t$, Memory Buffer $\mathcal{D}_m$, Maximum buffer size $\mathcal{M}$, Number of seen samples $\mathcal{N}$, Current sample $\{x, y\} \in \mathcal{D}_t$
**if** $\mathcal{M} > \mathcal{N}$ **then**
    $\mathcal{D}_m[\mathcal{N}] \leftarrow \{x, y\}$
**else**
    $i = randomInteger(min = 0, max = \mathcal{N})$
    **if** $i < \mathcal{M}$ **then**
        $\mathcal{D}_m[i] \leftarrow \{x, y\}$
**return** $\mathcal{D}_m$

---

## A.8 Hyperparameters

Table 6 provides the best hyperparameters used to report the results in Table 1. In addition to these hyperparameters, we use a standard batch size of 32 and a minibatch size of 32 for all our experiments.

Table 6: The best hyperparameters for IMEX-Reg to reproduce the results reported in Table 1

| Dataset | Buffer | LR | Epochs | $\gamma$ | $\eta$ | $\alpha$ | $\beta$ | $\lambda$ |
|---|---|---|---|---|---|---|---|---|
| Seq-CIFAR10 | 200 | 0.03 | 50 | 0.4 | 0.999 | 0.1 | 0.1 | 0.3 |
| | 500 | 0.03 | 50 | 0.4 | 0.999 | 0.1 | 0.2 | 0.3 |
| Seq-CIFAR100 | 200 | 0.03 | 50 | 0.08 | 0.999 | 0.1 | 0.3 | 0.15 |
| | 500 | 0.03 | 50 | 0.08 | 0.999 | 0.1 | 0.2 | 0.15 |
| Seq-TinyImageNet | 200 | 0.03 | 20 | 0.1 | 0.999 | 0.1 | 0.1 | 0.2 |
| | 500 | 0.03 | 20 | 0.15 | 0.999 | 0.1 | 0.1 | 0.3 |
| GCIL100 Uniform | 100 | 0.03 | 100 | 0.1 | 0.999 | 0.2 | 0.2 | 0.15 |
| | 200 | 0.03 | 100 | 0.1 | 0.999 | 0.2 | 0.2 | 0.15 |
| | 500 | 0.03 | 100 | 0.1 | 0.999 | 0.2 | 0.2 | 0.15 |
| GCIL100 Longtail | 100 | 0.03 | 100 | 0.1 | 0.999 | 0.2 | 0.2 | 0.15 |
| | 200 | 0.03 | 100 | 0.1 | 0.999 | 0.2 | 0.2 | 0.15 |
| | 500 | 0.03 | 100 | 0.1 | 0.999 | 0.2 | 0.2 | 0.15 |

## A.9 Hyperparamter tuning

Table 7 provides the results on hyperparameter tuning. As is evident from the results, our method is not very sensitive to the choice of hyperparameters.

Table 7: Hyperparamter tuning for IMEX-Reg on Seq-CIFAR100 with buffer size 200. As can be seen, IMEX-Reg is quite robust to the choice of hyperparameters.

| Varying $\alpha$, for $\beta = 0.3$, $\lambda = 0.15$ | | Varying $\beta$, for $\alpha = 0.1$, $\lambda = 0.15$ | | Varying $\lambda$, for $\alpha = 0.1$, $\beta = 0.3$ | |
|---|---|---|---|---|---|
| $\alpha$ | Top-1 Acc % | $\beta$ | Top-1 Acc % | $\lambda$ | Top-1 Acc % |
| 0.05 | 47.72 | 0.1 | 48 | 0.1 | 48.07 |
| **0.1** | **48.54** | 0.2 | 47.95 | **0.15** | **48.54** |
| 0.2 | 47.90 | **0.3** | **48.54** | 0.2 | 47.4 |
| 0.3 | 47.26 | 0.4 | 48.06 | | |

## A.10 Stability-Plasticity Trade-off

While forgetting $F_T$ quantifies how much knowledge is preserved in the model, we need to evaluate how well it is adapting to novel tasks. The extent to which CL systems need to be plastic in order to acquire novel information and stable in order to retain existing knowledge is known as the stability-plasticity dilemma. There is an inherent trade-off between the plasticity and stability of the model, and estimating this trade-off can shed some light on this dilemma. Sarfraz et al. (2022) proposes a *Trade-off* measure that approximates the balance between the stability and plasticity of the model. After learning the final task $T$, the stability ($S$) of the model is estimated as the average performance of all previous $T - 1$ tasks; $S = \sum_{i=0}^{T-1} A_{Ti}$.

The plasticity ($P$) of the model is measured as the average performance of each task after learning for the first time; $P = \sum_{i=0}^{T} A_{ii}$.

To find an optimal balance between the stability and plasticity of the model, Sarfraz et al. (2022) computes the harmonic mean of $S$ and $P$ as a trade-off measure, that is,

$$Trade\text{-}off = \frac{2SP}{S + P} \tag{11}$$

