# OpenReview forum: "IMEX-Reg: Implicit-Explicit Regularization in the Function Space for Continual Learning"
_TMLR — Accepted by TMLR_

### Review · Reviewer_Ptwz · 2024-03-01

**Summary Of Contributions:**

This paper proposes a method for continual learning that combines aspects of experience replay, contrastive representation learning, functional regularization and model averaging. It is shown that the resulting method is able to obtain higher performance than current replay-based methods on a number of relatively standard continual learning benchmarks.

**Audience:**

No

**Broader Impact Concerns:**

No concerns in this regard.

**Claims And Evidence:**

No

**Requested Changes:**

I would encourage the authors to focus on generating new insights and/or on demonstrating practical value of their engineered method.

Currently, it seems that the main contribution of this paper is getting a new “high score” on popular, but highly synthetical continual learning benchmarks (e.g. Split CIFAR). However, given the many degrees of freedom in these benchmarks (e.g. no control for computational complexity, no control for number of hyperparameters, no control for how many times the benchmark is tried), the value of such a new “high score” is very limited. (Moreover, it is very hard if not impossible for reviewers to really judge whether it indeed is a new high score to start with.) Rather, the value of these standard continual learning benchmarks is that they can be used for generating new insights by e.g. performing carefully controlled comparisons, but I’m afraid this is not what the current paper does.

Another avenue this paper could take is demonstrating practical value of the engineered method on actual real-world problems. The paper appears to attempt to claim that “generalized class-IL” provides such a real-world problem, but that is certainly not the case as this is simply a specific instance/variation of incremental CIFAR.

**Strengths And Weaknesses:**

An appealing aspect of this paper is that the motivation for including each of the different components of the proposed continual learning is quite extensively described. Generally, the paper is quite well written, although one critical note in this regard is that the paper is rather verbose. The paper has the tendency to describe things in an overly complex way, which can make it hard for the reader to understand the basics or underlying concepts, which often are quite straight-forward.

My main criticism of this paper is that, unfortunately, I do not think that it provides any new insights. After reading this paper, I’m afraid I don’t think I learned anything.
The main contribution of the paper is combining various established approaches to continual learning together (experience replay, contrastive representation learning, functional regularization and model averaging), and demonstrating that the resulting combined method performs better than current replay methods that use only some of these approaches. Doing this has probably required a decent amount of engineering effort to get everything to work, but the final result is not surprising or insightful. Moreover, given that the benchmark problems that are used in this paper are all highly synthetical and far away from any real-world applications, there is also virtually no practical value to this engineering effort.

I’m therefore afraid that I don’t think that this paper makes a contribution to the continual learning literature.

---

> ### Author Response · Authors · 2024-03-19
> **Response to Reviewer Ptwz**
>
> We express our gratitude to the reviewer for the encouraging words and the comprehensive review. Kindly find our response provided below:
>
> > My main criticism of this paper is that, unfortunately, I do not think that it provides any new insights.…..
>
> We respectfully disagree with the reviewer's assessment that the paper does not provide any new insights and is merely an engineering effort to get everything to work. As acknowledged by us both in the paper, Contrastive Representation Learning (CRL) and consistency/representation regularization using Exponential Moving Average (EMA) have been previously used in the CL literature. There are works that employ consistency regularization, such as CLS-ER [1], and both consistency regularization and CRL, such as OCDNet [2]. We compare and contrast such methods (CLS-ER and OCDNet) in Tables 1 and 2. As can be seen from our empirical evaluation, IMEX-Reg clearly outperforms these methods across different scenarios.
>
> Our novel contribution mainly lies in leveraging desirable traits of learning in the unit hypersphere through explicit classifier regularization in the function space. To the best of our knowledge, there are no works that leverage traits of learning in the unit hypersphere in CL. To this end, we align the geometric structures within the classifier hypersphere with those of the projection head hypersphere to compensate for weak supervision under low buffer regimes. Both through empirical evaluation (Tables 1 and 2) and also through an ablation study (Table 3), we highlight the effect of our novel contribution on several benchmark datasets. In addition, Section 5 on model characteristics uncovers additional benefits emanating from our novel contribution.
>
> We respectfully disagree with the reviewer's assertion that using synthetic benchmarks renders our findings in continual learning (CL) devoid of practical value. It's important to note that we have not introduced new benchmarks in CL; rather, we have utilized well-established benchmarks such as Seq-CIFAR10, Seq-CIFAR100, and Seq-TinyImageNet. These benchmarks were purposely designed to facilitate rapid prototyping of CL methods, taking into account computational and memory constraints. These benchmarks serve as important tools for initial validation and method development, allowing researchers to explore and refine novel approaches in a controlled setting before transitioning to more complex real-world applications. Therefore, construing the use of these benchmarks as a basis for dismissing the practical value of our research would inadvertently call into question the broader body of CL literature and its contributions to advancing the field.
>
> > I would encourage the authors to focus on generating new insights and/or on demonstrating practical value of their engineered method…..
>
> We respectfully disagree with the reviewer that the main contribution of our manuscript  lies in achieving the new ‘high score’. To the best of our knowledge, correlating inductive biases (especially regularization) with generalization has not been well studied in CL. Inspired by how humans leverage inductive biases, our main contribution lies in showcasing the efficacy of multitude of regularization strategies in CL. To this end, we leverage existing strategies such as Multi-task learning (implicit), consistency regularization using exponential moving average (explicit) and a novel regularization strategy to guide the classifier toward the activation correlations in the unit-hypersphere of the CRL. We also provide theoretical insights on feature similarity between CRL and cross-entropy loss, and Johnson-Lindenstrauss (JL) lemma connecting our novel loss to support our design decisions better. Furthermore, we show that such an approach  is robust to natural and adversarial perturbations, and mitigates task recency bias. Finally, due to IMEX-Reg’s design, unlabeled can also be leveraged to further improve generalization performance across tasks in CL. Although achieving a new ‘high score’ plays an important consideration in evaluating the contemporary methods,  our manuscript entails a multitude of analysis such as forgetting analysis, stability-plasticity trade-off, model calibration, task recency bias and robustness to corruptions to uncover the effectiveness of our contributions.
>
> In our current scope, we evaluate IMEX-Reg and contemporary methods on well established benchmarks. Although an evaluation on real-world problems holds significance, it is unfortunately beyond the scope of this paper.  We kindly request the reviewer to provide specific instances of real-world problems to see if we can accommodate it in our manuscript.

---

> > ### Comment · Reviewer_Ptwz · 2024-04-01
> >
> > Thank you for your rebuttal. Please find below some further comments that I hope will be useful.
> >
> > **Our novel contribution mainly lies in leveraging desirable traits of learning in the unit hypersphere through explicit classifier regularization in the function space.**
> >
> > I think that investigating the “leveraging of desirable traits of learning in the unit hypersphere” for continual learning is potentially an interesting direction. However, I’m afraid that I don’t think that the current paper provides clear and convincing evidence about the benefits of “leveraging desirable traits of learning in the unit hypersphere” for continual learning. Firstly (although this is a relatively minor point), it is not clear from the current version of the manuscript that this should be considered as the main contribution. Secondly, the motivation as well as the implementation details of this part of the method are not very clear, as also commented on by the other reviewers. Thirdly, the empirical effect / contribution of learning in the unit hypersphere is not made clear, and its contribution is never directly tested. The tables that the authors put forward in their rebuttal (Tables 1, 2 and 3) do not assess the contribution of this novel element; these tables only assess the performance of the combined method. Similarly, and contrary to what the authors seem to suggest in their rebuttal, it is also not shown that the model characteristics discussed in Section 5 are due to, or even related to, the “leveraging of desirable traits of learning in the unit hypersphere”.
> >
> > Although not mentioned in the rebuttal, it seems to me that the only experiment in which the contribution of this novel element is attempted to be tested directly, is the ablation study of Table 4. In this table it is however reported that the effect of the loss term that encourages learning in the unit hypersphere (i.e., $\mathcal{L}\_{\text{ecr}}$) is rather small. Indeed, even without this loss term the method used in this paper seems to outperform all tested baselines in Table 1! (This thus seems to indicate that the combination of various existing approaches is more important than the novel element introduced by this paper.) Moreover, it is unclear to me how this ablation experiment is performed exactly. For example, it seems to me that this experiment might have been performed by simply removing the component $\mathcal{L}\_{\text{ecr}}$. However, a better way to test this would be replacing this loss term with a “regular loss term” that is similar in every aspect except “encouraging learning in the unit hypersphere”.
> >
> > If the authors believe that “leveraging desirable traits of learning in the unit hypersphere” has important benefits for continual learning, I encourage them to try to address the issues raised above. In particular, I would encourage them to try to convincingly demonstrate this in the simplest possible setting rather than on top of a complex combined method, to remove possible confounds as much as possible. I would be interested to read such a paper, and I would be happy to review it again.
> >
> > **We kindly request the reviewer to provide specific instances of real-world problems to see if we can accommodate it in our manuscript.**
> >
> > I don’t think it is my place to come up with practical problems that your method could be useful for. Discussing such potential practical applications of the proposed method could however increase the value of the paper.

---

> > > ### Author Response · Authors · 2024-04-10
> > > **Reply to reviewer Ptwz**
> > >
> > > We are grateful for the comprehensive feedback provided by the reviewer. While we appreciate their input, we respectfully disagree with the notion that the empirical analysis presented in Tables 1, 2, and 3 does not effectively assess the impact of the novel explicit regularization proposed in this work. In order to ensure fairness in our comparisons, we adhered to well-established conventional benchmarks to evaluate the effectiveness of our method in continual learning (CL). Furthermore, we meticulously selected competing methods to thoroughly examine different aspects of our proposed approach. For instance, CLS-ER incorporates exponential moving average (EMA), while OCDNet relies on both EMA and contrastive representation learning (CRL). Taking this a step further, IMEX capitalizes on the desirable characteristics of learning in the unit hypersphere through explicit classifier regularization in the function space. Through extensive comparisons with CLS-ER and OCDNet, we aim to showcase the impact of our novel regularization on learning in CL.
> > >
> > > It is important to clarify that our primary objective was to introduce a new CL method that heavily relies on inductive biases to better manage the stability-plasticity trade-off. The main focus of this paper was not to establish a new, more realistic benchmark, but rather to propose a new CL method.

---

### Review · Reviewer_vQkA · 2024-03-04

**Summary Of Contributions:**

- This paper presents IMEX-Reg, a regularization method for continual learning that leverages both implicit regularization through an auxiliary loss using contrastive representation loss and explicit regularization using exponential moving average to enforce consistency. It also includes another regularization term that penalizes the differences between the unit normalized representations from the projection layer and the unit normalized representations from the classifier's projection layer.
- The approach shows strong performance across all Task-IL, Class-IL, and GCIL settings, outperforming baselines from recent work except for OCDNet on the Seq-CIFAR10 dataset. IMEX-Reg gets a low forgetting rate, is more robust to adversarial attacks or noise, and is better calibrated.

**Audience:**

Yes

**Broader Impact Concerns:**

There are no specific concerns for broader impact for this work.

**Claims And Evidence:**

Yes

**Requested Changes:**

- There is an overall need for improvement in clarity of the main narrative.
- Clarify how the projection layer $h$ is initialized and updated. Section 3.1 seems to indicate that $h$ is randomly initialized as an arbitrary 128-dimensional L2 normalized projection, and then the stopgrad term in Section 3.2 seems to indicate that $h$ is not updated. Also, what is the dimension size for the classifier projection $g$ ? Figure 1 seems to indicate that it's smaller than that of $h$.
- Why is $L_{cr}^{h}$ and $L_{cr}^{g}$ only applied to $D_{m}$ but not $D_{t}$ in Algorithm 1? Wouldn't applying the same terms when training with samples from $D_{t}$ also help with making sure that the model trained from $D_{1,\ldots,t-1}$ remain consistent?
- The introduction can contain more specifics with regards to the contributions to make it clearer earlier that the approach was tested on image classification datasets and at a high-level what the theoretical insights are.
- In the introduction, the motivation for studying CL with small buffer sizes mention privacy concerns, but this seems misplaced. Unless we use a federated learning setting, isn't privacy concern applicable to general ML, not confined to low-buffer regime CL? Also, it is unlikely that CL is performed on edge devices. A better motivation for low buffer CL is possibly when the number of tasks is large and each task has a lot of data such that training with new data even with a memory buffer becomes impractical, but I'm not sure whether the tasks explored in this paper correctly evaluate these setups.
- What is a low-buffer regime as opposed to a high-buffer regime? Is it a certain percentage of each task data or an absolute number or an absolute memory constraint?

**Strengths And Weaknesses:**

- Strengths:
	- The paper presents thorough quantitative analysis of their approach compared to a comprehensive set of baselines and settings using various analyses pertinent to continual learning. IMEX-Reg outperforms most baselines in the presented analyses.
	- The authors present an ablation study that validates the value of each component of IMEX-Reg.
- Weaknesses
	- While the results are strong, the benefit of the approach overall is confusing.
	- It is unclear how the projection layer $h$ , despite being randomly initialized and not updated through training, provides meaningful signal for effective regularization. Despite being randomly initialized and not updated, is the output from $h$ still able to capture the representation from Backbone $f$ ? Why can't we use the output directly from $f$ for $L_{cr}^{h}$ , $L_{rep}$ and apply L2 normalization for $L_{ecr}$ ? The output from $h$ contributes to all of the regularization terms except $L_{cr}^{g}$ , so it is important that this part is clarified.
	- Based on the above, it's unclear how the main contribution, guiding the classifier's projections toward the activation correlations in the unit-hypershere of the contrastive representation loss, leads to improvements.
	- It is unclear how IMEX-Reg is "inspired by how humans leverage inductive biases". Is it that humans needing minimal adaptations to learn new tasks mapped to these regularization terms? I don't think enough is known about the human learning process or at least it's not elaborated enough with related work in this paper to make this claim.
	- Without clearly defining low-buffer regime vs high-buffer regime and providing strong practical motivations for why studying the low-buffer regime is important, readers would be interested in how IMEX-Reg performs in a high-buffer regime compared to baselines and whether it still provides comparative gains.

---

> ### Author Response · Authors · 2024-03-19
> **Response to Reviewer vQkA (1/3)**
>
> We thank the reviewer for the encouraging words and the detailed review. Please find our response as follows:
>
> > While the results are strong, the benefit of the approach overall is confusing.
>
> IMEX-Reg addresses catastrophic forgetting in continual learning (CL) through a two-pronged implicit-explicit regularization strategy. IMEX-Reg leverages hard parameter sharing and multi-task learning to implicitly regularize the model. In addition, we introduce explicit regularization in the function space, guiding the optimization of the CL model towards generalization by leveraging desirable learning traits in the unit hypersphere. This novel regularization strategy encourages the classifier to align with activation correlations in the unit-hypersphere of the Contrastive Representation Learning (CRL) framework. The benefit of IMEX-Reg lies in its ability to effectively mitigate catastrophic forgetting in CL, improve generalization performance, and achieve robustness across natural and adversarial perturbations. We believe that our theoretical insights, combined with the empirical results, support the significance and effectiveness of our proposed approach.
>
> To further improve the paper's overall message, please let us know your suggestions on which sections or aspects of the paper would benefit from further clarification to convey the significance of the method.
>
> > Clarify how the projection layer is initialized and updated..
>
> We regret the lack of clarity with regard to the projection head in our method. Our CL model $ \Phi_\theta = \{ f, g, g^{'}, h \} $  consists of a shared backbone $f$, a linear classifier $g$, an MLP classifier projection $g^′$ , and a projection head $h$. The entire network is randomly initialized including the projection head. The CRL involves highly correlated multiple augmented views of the same sample which are propagated forward through the encoder f and the projection head h. Through Eqn. 2 and Eqn. 4, the projection head is updated in every iteration. However, we do not let gradients flow back through the projection head as we intend to distill knowledge from CRL to classification. It is a common practice within CRL literature to employ a non-linear projection head after CNN backbone as it improves representation quality (e.g. [1]).
>
> Therefore, we employ a learnable projection head to provide more flexibility for our model to learn meaningful representations which can then be leveraged for classification through our novel loss. We have updated our paper to reflect these clarifications.
>
> > Based on the above, it's unclear how the main contribution, guiding the classifier's projections toward the activation correlations in the unit-hypersphere of the contrastive representation loss, leads to improvements.
>
> The novelty of our method mainly lies in leveraging desirable learning traits in the unit hypersphere through explicit classifier regularization in the function space. Through a learnable backbone and projection head, we expect CRL to capture the global relationship between samples using instance discrimination tasks. We then seek to align the geometric structures within the classifier hypersphere with those of the projection head hypersphere to compensate for the weak supervision under low buffer regimes.
>
> > It is unclear how IMEX-Reg is "inspired by how humans leverage inductive biases"...
>
> Learning is easier when we have prior knowledge about the kind of problems that we will have to solve. Inductive biases are a means of embedding such prior knowledge into an optimization system. In the brain, CL is mediated by a plethora of neurophysiological processes that harbor strong inductive biases to encourage learning generalizable features, which require minimal adaptation when encountered with novel tasks. Importantly, the inductive biases that exist in the brain have been shaped by evolution to increase an animal’s fitness in the broad context of life on Earth [2]. It is worth noting that (i) many species, especially humans, develop slowly with large quantities of experiential data and (ii) that deep neural networks can work well in low data regimes if they have good inductive biases [3].
> Regularization has traditionally been used to introduce inductive bias in DNNs to prefer some hypotheses over others and promote generalization. Therefore, we seek to incorporate inductive biases in our CL model through implicit-explicit regularization. We have added more clarification in the related works to provide better understanding for the readers.

---

> > ### Author Response · Authors · 2024-03-19
> > **Response to Reviewer vQkA (2/3)**
> >
> > > Without clearly defining low-buffer regime vs high-buffer regime and providing strong practical motivations for why studying the low-buffer regime is important…..,
> >
> > While we acknowledge the significance of clearly defining and motivating the study of the low-buffer regime, it's important to note that the determination of low versus high buffer is often tied to the sample-to-task ratio. The low-buffer regime is characterized by a small sample-to-task ratio, signifying limited data availability per task, while the high-buffer regime entails a large sample-to-task ratio, indicating ample data per task, influencing the model's ability to handle continual learning challenges. However, it's essential to highlight that a specific threshold for this ratio may not be universally applicable, as it can vary based on the complexity of tasks, the nature of the data, and the particular learning scenario under investigation.
> > In our work, we explicitly address the low-buffer regime as it poses significant  challenges for CL systems, such as higher susceptibility to catastrophic forgetting and the need for effective memory consolidation and utilization.  We updated the related works to add more explanation on low-and high-buffer regimes.
> >
> > > There is an overall need for improvement in clarity of the main narrative.
> >
> > We have tried our best to incorporate all the changes suggested by you and other reviewers in Introduction, Related works and Method sections. Please let us know if any specific sections / topics need further revision to enhance the readability of the paper.
> >
> > > Clarify how the projection layer is initialized and updated….
> >
> > We have updated our Method section to provide more clarity on the projection head.
> >
> > > Why consistency regularization is only applied to memory samples but not to current task samples in Algorithm 1………
> >
> > When the current task introduces substantially different data distributions or necessitates divergent predictions compared to previous tasks, enforcing consistency with prior knowledge may not be advantageous and could potentially impede the learning of the new task. Moreover, the risk of catastrophic forgetting, wherein the model forgets how to perform well on previous tasks, is a significant concern. In such cases, consistency regularization to current task samples might introduce unnecessary complexity and hinder the learning process. In line with previous works (CLS-ER, OCDNet), we enforce consistency in predictions only for memory samples.
> >
> > > The introduction can contain more specifics with regards to the contributions to make it clearer earlier that the approach was tested on image classification datasets and at a high-level what the theoretical insights are.
> >
> > We have updated our Introduction to reflect these suggestions.
> >
> >
> > > In the introduction, the motivation for studying CL with small buffer sizes mentions privacy concerns, but this seems misplaced……
> >
> > We agree with the reviewer that privacy concerns are not uniquely confined to low-buffer regime continual learning. Therefore, we have removed the reference to privacy concerns from our manuscript to ensure clarity and alignment with the core motivations of our study. However, we respectfully disagree that CL is unlikely to be  performed on edge devices. On-device training for personalized learning is a challenging research problem [4]. The need to rapidly adapt deep prediction models at the edge is essential for catering to individual user requirements. Nevertheless, adapting models at the edge raises concerns regarding both the efficiency and sustainability of the learning process, as well as the ability to operate effectively under changing data distributions while coping with limited memory resources. Therefore, we argue that CL on the edge holds an important consideration in our design process to be efficient in buffered memory management.
> >
> > Additionally, we concur with the significance of the sample-to-task ratio in guiding continual learning, especially as the number of tasks increases, impacting the learning process even with a memory buffer. We recognize the importance of this factor in evaluating the challenges and dynamics of continual learning in various settings. In order to effectively evaluate models under a low buffer regime, we resort to well adopted buffer sizes in the literature: 100, 200, and 500. WIth highly limited buffer sizes in place, CL approaches overfit on the memory data and struggle to generalize well. IMEX-Reg improves generalization performance through a combination of implicit and explicit regularization techniques.
> >
> > > What is a low-buffer regime as opposed to a high-buffer regime? Is it a certain percentage of each task data or an absolute number or an absolute memory constraint?
> >
> > Please find the answer to this specific question above. We have updated the paper to provide more clarity on low-and high buffer regimes.

---

> > > ### Author Response · Authors · 2024-03-19
> > > **Response to Reviewer vQkA (3/3)**
> > >
> > > References:
> > >
> > > 1. Chen, Ting, et al. "Big self-supervised models are strong semi-supervised learners." Advances in neural information processing systems 33 (2020): 22243-22255.
> > > 2. Richards, Blake A., et al. "A deep learning framework for neuroscience." Nature neuroscience 22.11 (2019): 1761-1770.
> > > 3. Snell J, Swersky K, Zemel R. Prototypical networks for few-shot learning. 2017:4077–4087.
> > > 4. Pellegrini, Lorenzo, et al. "Continual learning at the edge: Real-time training on smartphone devices." arXiv preprint arXiv:2105.13127 (2021).

---

> > > > ### Comment · Reviewer_vQkA · 2024-03-20
> > > > **Response to rebuttal**
> > > >
> > > > Thank you for addressing my questions, I understand the work much better now.
> > > >
> > > > > Without clearly defining low-buffer regime vs high-buffer regime and providing strong practical motivations for why studying the low-buffer regime is important, readers would be interested in how IMEX-Reg performs in a high-buffer regime compared to baselines and whether it still provides comparative gains.
> > > >
> > > > I'm curious as to know what your thoughts on how IMX-Reg would work in a high-buffer regime. If it's generally useful for CL, would it still do well by using all available data for CL and have comparative gains compared to the works you compare?

---

> > > > > ### Author Response · Authors · 2024-03-21
> > > > > **Response to reviewer vQkA**
> > > > >
> > > > > We express our gratitude for the quick response from the reviewer. Here is our response:
> > > > >
> > > > > Continual Learning (CL) presents a significant challenge when data related to previous tasks diminishes or vanishes as the model progresses to the next task. With a limited buffer size, the CL model tends to learn features specific to buffered samples rather than representative features that are class- or task-wide. This often leads to poor performance on previously seen tasks. Consequently, various methods have been proposed to address this issue, including techniques such as consistency regularization [1], complementary learning systems [2], augmented rehearsal [3], and contrastive representation learning [4], all aimed at balancing the stability-plasticity trade-off.
> > > > >
> > > > > Conversely, in scenarios with a high buffer size, a plethora of samples corresponding to previously seen tasks are available. In such cases, CL models tend to suffer less from the stability-plasticity trade-off and demonstrate better generalization across tasks. Moreover, a very high buffer size effectively mirrors training a CL model jointly on all tasks at all times. In such scenarios, additional techniques to address the stability-plasticity trade-off become less impactful, as the trade-off itself is minimal. Empirical results, as observed in DER++ [1] and CLS-ER [2], indicate that the performance gap between simple experience rehearsal (ER) and more advanced methods diminishes as the sample-to-task ratio increases. Therefore, in scenarios with very high buffer ratios, we anticipate that the impact of regularization will decrease, and we expect IMEX-Reg to perform comparably with the majority of contemporary methods in experience rehearsal.
> > > > >
> > > > > References:
> > > > >
> > > > > [1]: Pietro Buzzega, Matteo Boschini, Angelo Porrello, Davide Abati, and Simone Calderara. Dark experience for general continual learning: a strong, simple baseline. In H. Larochelle, M. Ranzato, R. Hadsell, M. F. Balcan, and H. Lin (eds.), Advances in Neural Information Processing Systems, volume 33, pp. 15920–15930. Curran Associates, Inc., 2020
> > > > >
> > > > > [2] Elahe Arani, Fahad Sarfraz, and Bahram Zonooz. Learning fast, learning slow: A general continual learning method based on complementary learning system. In International Conference on Learning Representations, 2022.
> > > > >
> > > > > [3] Zhen Wang, Liu Liu, Yiqun Duan, and Dacheng Tao. Continual learning through retrieval and imagination. In AAAI Conference on Artificial Intelligence, volume 8, 2022a.
> > > > >
> > > > > [4] Jin Li, Zhong Ji, Gang Wang, Qiang Wang, and Feng Gao. Learning from students: Online contrastive distillation network for general continual learning. In Lud De Raedt (ed.), Proceedings of the Thirty-First International Joint Conference on Artificial Intelligence, IJCAI-22, pp. 3215–3221. International Joint Conferences on Artificial Intelligence Organization, 7 2022

---

### Review · Reviewer_HQzg · 2024-03-05

**Summary Of Contributions:**

The paper proposes a new approach to offline Class-IL and Task-IL, for replay based approaches where the buffer size is limited. The solution consists of a carefully crafted combination of losses designed to regularize the model and limit overfitting on the buffer. Concretely, the approach augments standard replay with the following losses :
1. $\mathcal{L}_{rep}$, a supervised contrastive objective on both new and buffered datapoints
2. $\mathcal{L}_{cr}$ a loss regularizing the classifier and projection head outputs to stay close to outputs of an EMA model
3. $\mathcal{L}_{ecr}$ ensuring that the similarities across points in the batch computed for both classifier head and projection heads align.

The authors evaluate their method on several well-established CL benchmarks and compare against strong baselines, showing that their method matches or surpasses previous state-of-the-art methods.
The authors also provide extensive analysis and ablations of their method, shedding light on how the proposed method's properties.

**Audience:**

Yes

**Claims And Evidence:**

Yes

**Requested Changes:**

### Critical changes
1. An small analysis of the impact of their method on buffer overfitting
2. A convincing argument / experiment showing that their use of the limited buffered memory is efficient
3. A discussion about the computational cost of their method

### Other changes
Adressing the questions I raised above.

**Strengths And Weaknesses:**

### Strengths :
1. The empirical results are strong and convincing. The experimental section is well crafted and highlights well the proposed approach.
2. The authors do a good job in additional analyses. The forgetting analysis, the stability / plasticity section are well-designed. Specifically, the analysis on model calibration and task recency bias, issues known to plague prior approaches in CL, are convincing and provide good support to the proposed method.
3. The ablation section is concise and easy to follow.

### Weaknesses :
1. **Motivation**. The proposed method is introduced as a solution to limit overfitting in small buffer regimes. While this has been shown to be an issue in prior works, the authors provide no empirical evidence that this problem is addressed by their method. Moreover, the argument made that having the features lie on the unit-hypersphere is somewhat weak. I agree that for SSL one needs to normalize the features to stabilize training, but I disagree that unit-norm makes representations easier to linearly separate for a classification head.
2. **Memory Usage**. The paper presents results in settings where the buffer memory is quite limited. While the setting is relevant, I am not convinced by the author's use of this memory. A Resnet-18 has approx 11 million parameters, so the memory to store both EMA and standard model is about 2 models x 11 miliion params x 4 bytes / param = 88 Megabytes. Storing 200 cifar-100 samples takes 200 x 32 x 32 x 3 x 1 byte / pixel = 0.6 Megabytes. Given the impact of the buffer size on performance, it's unclear whether the proposed approach makes optimal use of this memory (and this is without counting the memory needed to store intermediate activations required for backprop)
3. **Compute**. Lastly, the authors have not discussed the computation cost of their method. For instance, how would other methods fare if instead of forwarding data through the EMA model, that compute budget was used differently ?



### Small comments and questions :
1. Notation : The notation of $g$ and $g'$ is a bit confusing, maybe try using $g_{lin}$ and $g_{mlp}$ ?
2. I am not sure to understand what the role of $g'$ is, as it seems to be only used $\mathcal{L}_{cr}$. Have you considered removing this network, and instead computing $\mathcal{G}_h$ with either the logits ? Also, could you provide intuition on why the stopgrad operator is used ? Why not backprop through both matrices ?
3. For equation 3, why not simply use a smaller decay factor instead of of sporadically performing the update ? Is it for computational efficiency ?

---

> ### Author Response · Authors · 2024-03-19
> **Reply to Reviewer HQzg (1/2)**
>
> We thank the reviewer for giving their valuable feedback on our paper. We appreciate the reviewer for acknowledging the additional analyses that often go overlooked in CL literature. We address the concerns raised by the reviewer below.
> > The argument made that having the features lie on the unit-hypersphere is somewhat weak…:
>
> We apologize for any misunderstanding concerning our rationale for aligning the feature representations within the unit hyperspheres. It is important to clarify that we do not assert that the utilization of unit norm inherently renders the feature representations linearly separable within the classifier decision space.
> Wang & Isola [1] highlights that representation learning in the unit hypersphere could result in linearly separable clusters, a widely used criterion for representation quality. Therefore, we hypothesize that the projection of the classification head onto a unit hypersphere, coupled with alignment with contrastive representation learning, could potentially guide the classifier's decision space to discern more effective boundaries between distinct classes. By leveraging the desirable traits of representation learning in the classifier, we show that the generalization performance can be greatly improved.
> We have updated section 3.2 to provide more clarity on the same.
> > Memory Usage
>
> IMEX-Reg effectively utilizes its buffered samples and attains comparable performance to other baselines, even when using significantly fewer samples. For instance, in the GCIL-Longtail scenario, IMEX-Reg surpasses DER++ with 1/5-th buffer size. However, in the context of the widely used CL benchmarks with relatively small image sizes, the advantage of reducing buffer size is overshadowed by the EMA model's size. Transitioning to more real-world settings, such as ImageNet, would reduce the disparity between the EMA model size and buffer size. For example, with 200 samples in the ImageNet dataset (assuming an image size of 256x256), the storage requirement would be 38.4 Megabytes, which is comparable to the EMA model size. As can be seen from Table 1 and Table 2, the performance gain of IMEX-Reg over other baselines increases as the complexity of the CL setting increases. Therefore, we hypothesize that IMEX-Reg will achieve similar performance compared to other baselines with a significantly lower number of buffer samples in extremely challenging scenarios like ImageNet. In such cases, the additional memory overhead from having an extra EMA model is much less compared to the number of samples saved from being stored in the buffer.
> > Authors have not discussed the computation cost of their method.
>
> Table below provides a relative comparison of training times for different CL approaches trained on Seq-CIFAR100 with a buffer size of 200. A large part of IMEX-Reg's; as well as OCDNet's training time can be attributed to computations involving CRL. Our novel contribution $\mathcal{L_{ecr}}$ adds up only a minimal computational overhead. Although IMEX-Reg takes longer time to train, the performance improvement is significant enough to sidestep computational overhead.
> During inference we discard the projection head, so IMEX-Reg’s processing time is the same as the other approaches
>
>  Method|DER++|CLS-ER|OCDNet|IMEX-Reg
> -|-|-|-|-
> Relative time taken | 1x | 1.09x | 1.44x | 1.57x
>
> We have added this discussion on the computation cost to the Appendix of the paper.
> > Notation
>
> We regret the confusion regarding g and g’. We have updated the notations in the paper as suggested.
> > I am not sure what the role of g’ is
>
> We use the MLP network, g’,  to transform the classifier decision space to a unit hypersphere for aligning the representations with the CRL head. By doing so, we do not restrict the classifier decision space to be a unit hypersphere. The classifier projection head will align with the CRL head and guide the classifier decision space to better boundaries. Below we provide the accuracy measures for IMEX-Reg with and without g’ MLP network for Seq-CIFAR100 with different buffer sizes for 3 seeds. As can be seen,  adding the MLP network improves the method’s generalization capabilities.
> Buffer size|Without g’|With g’
> -|-|-
> 200|46.14 (0.43)|48.54 (0.23)
> 500|54.90 (0.97)|56.53 (0.80)
> > Could you provide intuition on why the stopgrad operator is used ?
>
> Our intuition for using the StopGrad was to allow the CRL head to independently learn the generalizable representations without any other influence from the classifier. Constraining the CRL head representations to be similar to the classifier head’s projection representations could potentially lead to suboptimal learning in the CRL head. Therefore, we provide a one-way channel to drive the classifier’s decision space by aligning representations in its unit hypersphere with that of the CRL head.

---

> > ### Author Response · Authors · 2024-03-19
> > **Reply to Reviewer HQzg (2/2)**
> >
> > > why not simply use a smaller decay factor instead of of sporadically performing
> >
> > We sporadically update the EMA model parameters for multiple reasons. Firstly, this approach ensures that our experimental setup is consistent with other methods that utilize the EMA model, thereby enabling fair comparisons. Secondly, this sporadic updating reduces the overlap between the snapshots of the model, introducing greater diversity in the EMA model. Moreover, a stochastic rather than a deterministic approach is more biologically plausible[2]. Finally, as the reviewer pointed out, by updating the model intermittently, we can achieve computational savings
> > > An small analysis of the impact of their method on buffer overfitting
> >
> > We have focussed our paper on low buffer regimes where the sample per class ratios can be as low as 1 in several CL settings. Repeated learning on this limited data from past tasks, results in overfitting on buffered samples and hence catastrophic forgetting of past information. In such scenarios, experimental analysis in Table 1 and Table 2 shows that many other methods fail to learn generalizable representations from the limited data. However, IMEX-Reg encourages the model to learn generalizable representations through CRL and our novel explicit regularization. These performance improvements, especially in low buffer regimes, serve as an evidence to IMEX-Rex resilience to overfitting on limited buffer data.
> > If there are any pertaining doubts, we would greatly appreciate the reviewer's suggestion for any particular experiment they would like to see.
> > > A convincing argument / experiment showing that their use of the limited buffered memory is efficient
> >
> > In our experimental evaluation, we assessed various CL models using established benchmarks with increasing complexity. These benchmarks encompass a range of scenarios where the number of samples per class varies from 50 to 1, across different CL settings and buffer sizes. We observed that as the number of samples per class decreases, the generalization of methods tends to deteriorate, implying less effective utilization of buffer samples. However, IMEX-Reg consistently demonstrates superior performance even with a reduced number of buffer samples. For instance, the table below compares the performance of DER++ with a buffer size of 500 to IMEX-Reg with a buffer size of 200. As can be seen IMEX-Reg notably outperforms DER++ in nearly all scenarios, despite having half the buffer size. This enhancement in performance can be attributed to IMEX-Reg's ability to learn generalizable representations, thus effectively leveraging its buffer samples.
> >
> > Method| Seq-CIFAR10| Seq-CIFAR100| Seq-TinyImageNet| GCIL-Uniform| GCIL-Longtail
> > -|-|-|-|-|-
> > DER++: buffer size 500 |72.7| 41.4| 19.38|35.83|34.23
> > IMEX-Reg: buffer size 200|71.56|48.54|24.15|43.19|42.66
> > Acc improvements|-1.57%|17.25%|24.61%|20.54%|24.63%
> >
> > We have added this discussion to the Appendix to provide more clarity to the readers
> > > A discussion on computational cost
> >
> > We have already covered the question above.
> >
> > We once again thank the reviewer for the feedback and the appreciation of our empirical results. We believe we have addressed all the concerns raised and hope our clarifications improve the confidence of the reviewer in our paper. We are more than happy to address any pertaining issues the reviewer may have.
> >
> > 1. Tongzhou Wang and Phillip Isola. Understanding contrastive representation learning through alignment and
> > uniformity on the hypersphere. In International Conference on Machine Learning, pp. 9929–9939. PMLR, 2020.
> > 2. Wolfgang Maass. Noise as a resource for computation and learning in networks of spiking neurons. Proceedings of the IEEE, 102(5):860–880, 2014

---

> ### Comment · Reviewer_HQzg · 2024-04-05
> **Response to Rebuttal**
>
> Thank you for answering the questions raised in the review, and for the additional experiments to support the claims.
>
> Most of my points of concerns have been addressed. I am still unsure about the argument given by the authors on benefits on the unit-hypersphere. I understand that to evaluate representations learned in a self-supervised way, linear separability is a good property to have. That being said, in supervised settings this is optimized directly with a standard cross-entropy objective. That being said, I agree that features learned from an unsupervised loss could help with buffer overfitting (though I somewhat disagree that this is due directly to the features lying on the unit-hypersphere).
>
> My last point of concern is the author's claim that better performance directly implies less buffer overfitting. For me this may not directly be the case (e.g. you could have a mode that has perfect training accuracy, but still generalizes well to new points). I would suggest to the authors to either prove this fact directly (that buffer overfitting is reduced) or to reframe the motivation of the method.
>
> Thank you.

---

> > ### Author Response · Authors · 2024-04-09
> > **Reply to the reviewer HQzg**
> >
> > We thank the reviewer for engaging in further discussion on our manuscript. We are glad to have answered majority of your concerns. Our response to further questions are as follows:
> >
> > Several self-supervised learning approaches, especially contrastive learning in this case, learn representations with a unit-norm constraint, effectively restricting the output space to the unit hypersphere. This constraint brings forth several desirable traits, including training stability and plausible linear separability. As is the case with OCDNet, CRL shares the backbone with cross-entropy and acts as an implicit regularizer. Additionally, the aforementioned desirable characteristics can be further leveraged for explicit classifier regularization, thereby compensating for weak supervision under low buffer regimes. In Tables 1, 2, and 3, we empirically showcase the impact of such explicit regularization in CL. Please let us know if you have any suggestions on how we can clarify this confusion further in our manuscript.
> >
> > We concur with the reviewer that the inverse correlation between model performance and buffer overfitting might not be reasonable without additional supporting evidence. Therefore, we plan to include an experiment that showcases the effect of novel explicit regularization on the loss landscape of buffered samples in our final revision. We anticipate IMEX-Reg to retain maximum stability while committing minimum errors on the buffered samples.
> >
> > We have endeavored to address the questions raised to the best of our ability. Please let us know if any concerns remain. We are more than willing to engage in further discussion to address any remaining questions.

---

### Decision · Action_Editor_Rgm7 · 2024-04-20

**Recommendation:** Accept with minor revision

**Comment:**

This paper proposes to combine several existing techniques for continual learning to improve the performance, and this is shown to work on several benchmarks. The paper was reviewed by three reviewers. All reviewers acknowledged the empirical performance of the proposal. All reviewers, however, raised concerns around novelty of the proposal, which I agree with. Given that novelty is not an explicit decision-making factor in TMLR, and given that the paper meets the bar in terms of substantiation of the claims, it is recommended to be accepted subject to minor revisions. Congratulations!

- Please remove mentions of novelty claims in the paper and focus on factually describing the proposal.

- At the end of page 2, in contributions, please give references to where the each of the proposed components appears.

- The equations are generally missing punctuation, e.g., the unnumbered eq. in Theorem 2. Please check all equations for proper punctuation.

- Please make the code publicly available as promised.

- Please add clarity to the claim of "leveraging desirable traits of learning in the unit hypersphere through explicit classifier regularization in the function space."

**Audience:**

The paper is well within scope of TMLR.

**Claims And Evidence:**

The claims are largely supported by evidence in the paper. There are minor revisions that are still needed.

---

> ### Author Response · Authors · 2024-04-28
> **revised manuscript**
>
> We would like to thank all the reviewers and the Action Editor for their valuable feedback and time. We have incorporated all the suggestions and have made the code available online.